# Nascent-Seq reveals novel features of mouse circadian transcriptional regulation

**Jerome S Menet\*, Joseph Rodriguez, Katharine C Abruzzi, Michael Rosbash\***

Howard Hughes Medical Institute, National Center for Behavioral Genomics, and Department of Biology Brandeis University, Waltham, United States

**Abstract** A substantial fraction of the metazoan transcriptome undergoes circadian oscillations in many cells and tissues. Based on the transcription feedback loops important for circadian timekeeping, it is commonly assumed that this mRNA cycling reflects widespread transcriptional regulation. To address this issue, we directly measured the circadian dynamics of mouse liver transcription using Nascent-Seq (genome-wide sequencing of nascent RNA). Although many genes are rhythmically transcribed, many rhythmic mRNAs manifest poor transcriptional rhythms, indicating a prominent contribution of post-transcriptional regulation to circadian mRNA expression. This analysis of rhythmic transcription also showed that the rhythmic DNA binding profile of the transcription factors CLOCK and BMAL1 does not determine the transcriptional phase of most target genes. This likely reflects gene-specific collaborations of CLK:BMAL1 with other transcription factors. These insights from Nascent-Seq indicate that it should have broad applicability to many other gene expression regulatory issues.

## Introduction

Most organisms from bacteria to humans possess circadian rhythms, which generate oscillations in biochemistry, physiology and behavior. The circadian system in eukaryotes is based on cell-autonomous molecular oscillators, which rely on transcriptional feedback loops. In mammals, the transcription factor BMAL1 acts as a dimer with either CLOCK (CLK) or Neuronal PAS domain protein 2 (NPAS2) to activate the transcription of many genes, including the transcriptional repressors *Period* (*Per1*, *Per2* and *Per3*) and *Cryptochrome* (*Cry1* and *Cry2*). The PERs and CRYs are expressed, post-translationally modified, feedback to inhibit their own transcription and are then rhythmically degraded to lead to a new round of BMAL1:CLK or BMAL1:NPAS2 -mediated transcription (reviewed in *Ko and Takahashi, 2006*; *Dardente and Cermakian, 2007*). This temporal regulation of clock gene transcription cycles with a period of about 24 hr and probably underlies much of circadian biology.

Over the past decade, clock gene transcriptional regulation has been described in many species and tissues, where it drives the rhythmic expression of a large fraction of the mRNA population (up to 10–15% of all mRNAs in a single mammalian tissue; *Lowrey and Takahashi, 2004*; *Vollmers et al., 2009*). Rhythmic mRNA expression has mostly been characterized by analyzing temporal changes of steady-state mRNA levels, using techniques such as microarrays (e.g., *McDonald and Rosbash, 2001*; *Panda et al., 2002*; *Storch et al., 2002*) and more recently high-throughput sequencing (*Hughes et al., 2012*). It is generally assumed that these rhythms in mRNA expression directly result from temporal changes in transcription. There are, however, a few reports indicating that post-transcriptional regulation contributes to rhythmic mRNA expression of several genes, including core clock genes (reviewed in *Kojima et al., 2011*; *Staiger and Green, 2011*; *Staiger and Koster, 2011*; *Zhang et al., 2011*), but this has never been studied in detail at the genome-wide level. Circadian post-transcriptional regulation may impact rhythmic mRNA expression at many different levels, such as mRNA splicing, stability and translation. For example, post-transcriptional events rhythmically regulate the mRNA half-life of the mammalian clock genes *Per2* and *Cry1* and the *Drosophila* clock gene *per* (*So and Rosbash,*

**\*For correspondence:** rosbash@brandeis.edu (MR); menet@brandeis.edu (JSM)

**Competing interests:** The authors have declared that no competing interests exist

**Reviewing editor**: Todd C Mockler, Donald Danforth Plant Science Center, United States

**eLife digest** Many biological processes oscillate with a period of roughly 24 hr, and the ability of organisms as diverse as bacteria and humans to maintain such circadian rhythms, even under conditions of continuous darkness, influences a range of phenomena, including sleep, migration and reproduction. One characteristic of circadian rhythms is that they can adjust to local time (with humans suffering from jet lag as they wait for this to happen).

Experiments have shown that the circadian system in mammals relies on feedback loops that operate at the level of individual cells. These loops are controlled by two particular proteins, which comprise the transcription factor complex called BMAL1:CLK. Transcription factors cause particular sequences of bases in the DNA of cells to be transcribed into messenger RNA, thus starting the process by which target genes are expressed as proteins. In the case of BMAL1:CLK, these proteins are then modified, which inhibits any further transcription of the target genes. A reversal of these modifications is then followed by the synthesis of new proteins, which allows a new cycle of the transcription process to begin.

The amounts of many messenger RNAs (mRNAs) in a cell also increases and decreases with a period of 24 hr, and it was generally assumed that this was due to the changes in the level of transcription. More recently, however, it was suggested that other processes, such as splicing and translation, might also contribute to rhythmic changes in the amount of mRNA associated with particular genes. Such post-transcriptional processes are known to have a role in other areas of cell biology, including aspects of the circadian system, but until very recently this had not been studied in detail for all genes.

Now Menet et al. have directly assayed rhythmic transcription by measuring the amount of nascent mRNA being produced at a given time, six times a day, across all the genes in mouse liver cells using a high-throughput sequencing approach called Nascent-Seq. They compared this with the amount of liver mRNA expressed at six time points of the day. Although the authors found that many genes exhibit rhythmic mRNA expression in the mouse liver, about 70% of them did not show comparable transcriptional rhythms. Post-transcriptional regulation must, therefore, have a major role in the circadian system of mice and, presumably, other mammals.

Menet et al. also found that the influence of CLK:BMAL1 differed from what was expected, which suggests that it collaborates with a number of other transcription factors to effect transcription of most target genes. In addition to showing that circadian systems of mammals are more complex than previously believed, the results also illustrate the potential of Nascent-Seq as a genome-wide assay technique for exploring a range of questions related to gene expression and gene regulation.

*1997*; *Woo et al., 2009*; *Woo et al., 2010*). Moreover, several RNA-binding proteins such as LARK, hnRNP I, hnRNP P, hnRNP Q or the circadian deadenylase NOCTURNIN have been shown to regulate circadian gene expression and/or circadian behavior (reviewed in *Kojima et al., 2011*). These different modes of post-transcriptional regulation are not restricted to circadian biology (*Keene, 2007*) and have been shown in other systems to regulate cellular mRNA levels independent of transcriptional regulation (*Giege et al., 2000*; *Cheadle et al., 2005*).

To address the genome-wide contribution of transcriptional and post-transcriptional regulation to mammalian mRNA rhythms, we used Nascent-Seq (high-throughput sequencing of nascent RNA; *Carrillo Oesterreich et al., 2010*; *Khodor et al., 2011*) to assay global rhythmic transcription in mouse liver. We performed a parallel analysis of rhythmic mRNA expression with RNA-Seq and compared the two sequencing datasets. Although many genes are rhythmically transcribed in the mouse liver (~15% of all detected genes), only 42% of these rhythmically transcribed genes show mRNA oscillations. More importantly, about 70% of the genes that exhibit rhythmic mRNA expression do not show transcriptional rhythms, suggesting that post-transcriptional regulation plays a major role in defining the rhythmic mRNA landscape. To assess the contribution of the core molecular clock to genome-wide transcriptional rhythms, we also examined how rhythmic CLK:BMAL1 DNA binding directly affects the transcription of its target genes. Although maximal binding occurs at an apparently uniform phase, the peak transcriptional phases of CLK:BMAL1 target genes are heterogeneous, which

indicates a disconnect between CLK:BMAL1 DNA binding and its transcriptional output. The data taken together reveal novel regulatory features of rhythmic gene expression and highlight Nascent-Seq as an important genome-wide assay for the study of gene expression.

## Results

### Genome-wide analysis of transcription in the mouse liver using Nascent-Seq

To address the regulation of genome-wide transcription, we analyzed mouse liver nascent RNA expression, that is, RNA being transcribed by RNA Polymerase II (Pol II) prior to 3′ end formation (from 12 independent samples in LD, 6 time points per day done twice; see analysis of rhythmic transcription in mouse liver section). To this end, nascent RNA was extracted from purified nuclei using the high salt/urea/detergent buffer originally described by *Wuarin and Schibler (1994)*. A very similar sequencing strategy was recently applied by Smale, Black and colleagues to macrophage nascent RNA (*Bhatt et al., 2012*). We then prepared illumina libraries with standard protocols for high-throughput sequencing (Nascent-Seq; *Carrillo Oesterreich et al., 2010*; *Khodor et al., 2011*). Removal of rRNA was unnecessary as approximately 65–70% of the sequences uniquely mapped to the genome (*Table 1*).

Seventy six percent of these uniquely mapped sequences map to introns (*Figure 1A*). This contrasts dramatically with more conventional RNA-Seq; it has minimal intron reads as it assays polyadenylated (pA) RNA and therefore predominantly mature (spliced) mRNA (*Figure 1A*). Intronic Nascent-Seq reads are also more abundant in mouse compared to *Drosophila* (76% vs 45%), reflecting longer intron size and less efficient mouse co-transcriptional splicing (*Khodor et al., 2012*; *Khodor et al., 2011*). Many genes exhibit a 5′ to 3′ gradient in the Nascent-Seq dataset, presumably reflecting nascent RNAs of different lengths attached to elongating Pol II (*Figure 1B*). In addition, Nascent-Seq signals frequently extend past the polyadenylation site, reflecting RNA not yet cleaved by the cleavage/polyadenylation specificity factor (CPSF) and/or RNA molecules still associated with Pol II after cleavage but prior to degradation by the 5′ to 3′ exoribonuclease *Xrn2* (*Figure 1C*). These features are absent from standard RNA-Seq data, and indicate that Nascent-Seq predominantly detects nascent RNA molecules attached to elongating Pol II (*Figure 1B,C*).

Another feature was apparent in the comparison of Nascent-Seq and RNA-Seq: gene expression was often different between the two datasets. Some genes have a high ratio of Nascent-Seq to RNA-Seq signal (e.g., *B4galt1*, *Figure 1B*), whereas others have a low ratio (e.g., *Bag1*, *Figure 1B*). Genes with a high ratio (top 10% of all genes) are dramatically enriched for specific functions, namely, non-coding RNA (ncRNA), G-coupled protein receptors, regulation of transcription and chromatin organization (*Figure 1D,E*). Genes with a low ratio are enriched for genes involved in ribosome function and mitochondrial respiration (*Figure 1D,E*). Because these genes are associated with short or long mRNA half-lives, respectively, we compared the Nascent-Seq to RNA-Seq ratios of genes with their published mRNA half-lives (*Sharova et al., 2009*). Not surprisingly, genes with a high ratio have relatively short half-lives, and genes with low ratios have longer mRNA half-lives (*Figure 1F*). These data indicates that mRNA stability contributes to the wide range of Nascent-Seq to RNA-Seq ratios.

### Analysis of rhythmic transcription in mouse liver

To identify genes that are rhythmically transcribed, we performed two independent six time-points rhythms of mouse liver Nascent-Seq. We found that some genes exhibit very high amplitude rhythms, with no detectable signal at low time points (e.g., *Npas2*; *Figure 2A*). About 15% of expressed genes manifest transcriptional rhythms (p<0.05: 6.3%, strong rhythms; 8.9% medium-strength rhythms; see 'Materials and methods' for analysis details; *Figure 2B*). Phases of maximal transcription are heterogeneous yet not uniformly distributed; very few genes peak at ZT16-20 in the mid-late night (*Figure 2C,D*).

Protein-coding genes dominate the cycling Nascent-Seq dataset (>85%, data not shown), but it also contains genes encoding ncRNAs, for example, pri-miRNAs and long non-coding RNAs (lncRNA) (*Figure 2E,F*; *Figure 2—figure supplement 1–6*). As previously described (*Gatfield et al., 2009*), pri-miRNA 122a is robustly rhythmic (*Figure 2E*). However, rigorous quantitation of rhythmic ncRNA transcription is precluded by the poor annotation of these transcription units.

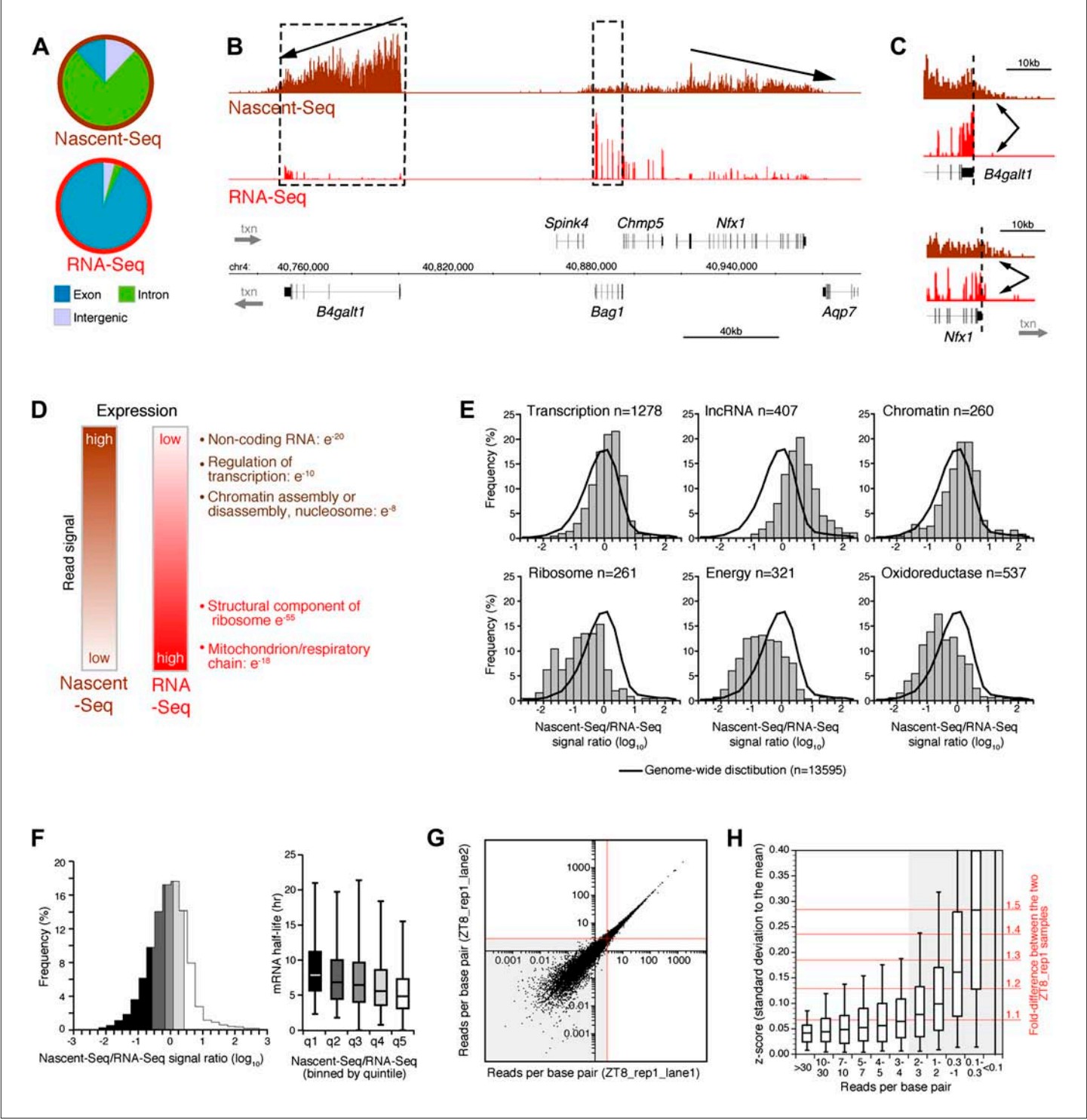

**Figure 1**. Genome-wide assay of transcription in the mouse liver using Nascent-Seq. (**A**): Distribution of high-throughput sequencing signal within introns (green), exons (blue) and intergenic regions (grey) for Nascent-Seq and RNA-Seq datasets. (**B**): Visualization of Nascent-Seq and RNA-Seq signal at chr4: 40,730,000–41,002,500. Genes above the scale bar are transcribed from left to right and those below the scale bar are transcribed from right to left. Nascent-Seq signal exhibits increased intron signal and a 5′ to 3′ gradient signal (arrow). Moreover, differences between Nascent-Seq signal and RNA-Seq signal are observed for many genes (e.g., *Bag1* and *B4galt1*). (**C**): Nascent-Seq signal (brown), but not RNA-Seq signal (red), extends past the annotated 3′end of the genes *B4galt1* and *Nfx1*. (**D**): Gene ontology of genes with high Nascent-Seq and low RNA-Seq signals (and inversely) is indicative of RNA with short or long half-lives, respectively (see 'Materials and methods' for details). (**E**): Distribution of the Nascent-Seq/RNA-Seq signal ratio for the classes of genes enriched in (**D**). (**F**): Nascent-Seq/RNA-Seq signal ratio significantly correlates with mRNA half-lives (values from

*Figure 1. Continued on next page*

*Figure 1. Continued*

**Sharova et al., 2009**), and genes with high ratio display shorter half-lives and inversely. (**G**) and (**H**): Strategy used to determine the gene signal cut-off threshold used in our analysis. Variation of gene signal coming from the sequencing of a Nascent-Seq library (G; ZT8, replicate 1) sequenced in two Illumina flow-cell lanes was assessed by calculating the z-score (**H**). Less than 5% of the genes with a read per base pair superior to three exhibit a 1.3-fold gene signal variation. See 'Materials and methods' for more details.

## Transcriptional rhythms overlap poorly with rhythms in mRNA expression

To address the relationship of cycling transcription to cycling mRNAs, we assayed two independent six time-points profiles of pA RNA by RNA-Seq. The comparison with Nascent-Seq was restricted to genes that were sufficiently expressed in both datasets (n = 5454 genes; see 'Materials and methods' for details). Using the identical statistical analysis and cut-offs to determine rhythmicity, the fraction of rhythmic mRNA was higher than the fraction of rhythmic nascent RNA (22.1% and 15.1%, respectively; *Figure 3A*).

There was also a notably poor overlap between the two rhythmic gene sets: only 41.6% of rhythmically transcribed genes also manifest rhythmic mRNA expression (R-R gene set; 342/822; *Figure 3A–C*). However, the mRNA phase of these R-R genes was highly correlated to the nascent RNA phase (r = 0.92; *Figure 3C*), and more than half (57%) of these genes exhibit a phase difference of less than 2 hr (195/342; *Figure 3D*). The amplitudes of the nascent RNA rhythms were also correlated with those of mRNA (r = 0.76; *Figure 3E*), indicating that transcriptional regulation dominates these R-R rhythms. Not surprisingly, almost all clock genes (*Figure 4*) and well-characterized clock-controlled genes (e.g., *Nocturnin, Por, Alas1, Upp2, Usp2, Inmt, Nfil3*, etc; *Figure 3—source data 1*) are in this R-R gene set.

The other 58.4% of rhythmically transcribed genes (480/822) do not show robust mRNA expression (R-AR gene set; *Figure 3A,F*). A simple explanation is that the mRNA half-lives of these R-AR genes are relatively long and therefore mask the transcriptional oscillations. However, these genes do not have altered nascent RNA to mRNA ratios compared to the whole genome (*Figure 5A*) or reported longer mRNA half-lives (assessed using the dataset from *Sharova et al., 2009*; *Figure 5B*). These considerations suggest that other mechanisms account for the poor mRNA oscillations of this gene set (e.g., the rhythmic Nascent-Seq signal of 25 R-AR genes results from rhythmic transcription of an adjacent gene that reads into the R-AR gene; *Figure 5C*).

The opposite comparison is based on genes with rhythmic mRNA expression, of which only 28.4% (342/1204) have rhythmic transcription that meets the cycling criteria (*Figure 3A,G*). This surprising conclusion was similar when the analysis was restricted to genes with the strongest mRNA rhythms (121/435) and indicates that most cycling mRNAs (862/1204) likely undergo post-transcriptional regulation. This might include the circadian regulation of nuclear RNA processing, export, translational regulation and/or mRNA turnover, as described for the few circadian genes shown to be regulated post-transcriptionally (*Kojima et al., 2011*; *Staiger and Koster, 2011*; *Zhang et al., 2011*). Gene ontology analysis of this arrhythmic transcription-rhythmic mRNA (AR-R) gene set did not reveal any striking enrichment of particular gene functions (*Figure 5D*).

## Transcriptional variability contributes to rhythmic mRNA expression

There is a peculiar feature of the large number of genes within this AR-R category: visualization of RNA expression with heatmap indicates that many of these transcriptionally 'arrhythmic' genes manifest elevated transcription at times that match their cycling mRNA peaks (*Figure 3G*; note that the heatmaps show that the transcription peak of many genes matches the peak phase of mRNA cycling). Further inspection of all individual expression profiles confirmed this correlation: many genes have matching peak phases despite large variations in nascent RNA expression (*Figure 6A*).

To substantiate this variability in nascent RNA expression, we calculated the standard deviation normalized to the mean (SD) for every gene using the two 6 time-points profiles as 12 independent samples. The reasoning was that low variability expression between time points should result in a low SD. The SD was noticeably higher in the Nascent-Seq dataset than in the RNA-Seq dataset, indicating that the transcription of most AR-R genes is indeed variable compared to the mRNA-Seq dataset (*Figure 6C*; note that most points are to the right of the diagonal line). Higher variability of nascent

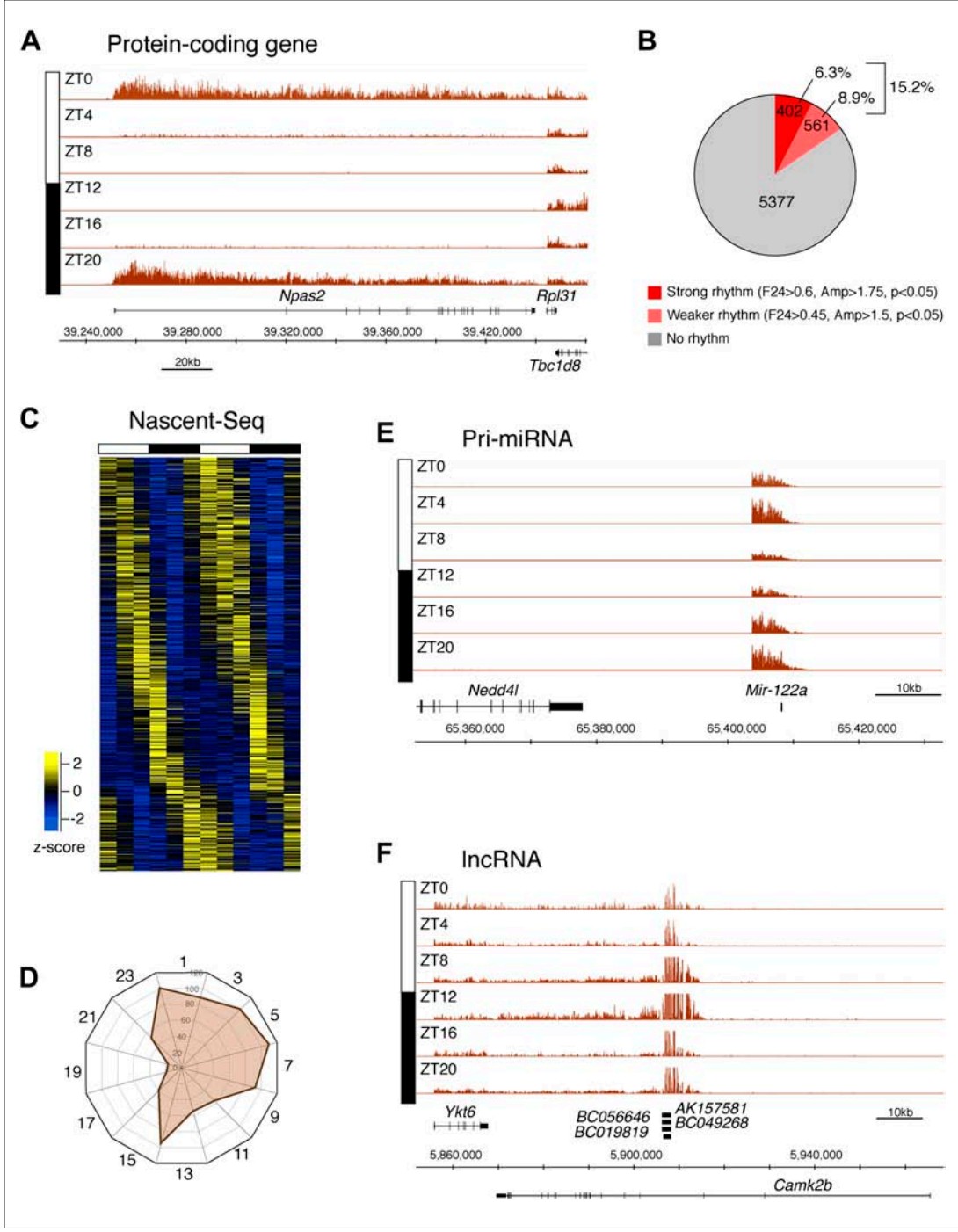

**Figure 2**. Genome-wide analysis of rhythmic transcription in the mouse liver. (**A**): Visualization of *Npas2* Nascent-Seq signal at six time points of the light:dark cycle (first replicate). *Npas2* Nascent-Seq signal is rhythmic and peaks at ZT20-ZT0, contrary to the signal within the adjacent gene *Rpl31*. (**B**): Quantification of the number of genes that are rhythmically transcribed in the mouse liver. Genes with more than three reads per base pair for at least one time point were included for the analysis. Genes are considered to be rhythmically transcribed if signal amplitude (Amp) is greater than 1.5, if signals for the 12 time points follow a sinusoid curve ($F_{24} > 0.45$) and if the $F_{24}$ value is in the top 5% of all $F_{24}$ values calculated after time points were permutated 10,000 times (p<0.05). A rhythm was considered to be strong (dark red) if $F_{24} > 0.6$ and Ampl > 1.75. (**C**): Heatmap representation of Nascent-Seq signal for the 963 genes that are rhythmically transcribed in the mouse liver. High expression is displayed in yellow (z-score > 1), low expression in blue (z-score < 1). (**D**): Expression phase of rhythmically expressed nascent RNA (n = 936) was separated by bins of 2 hr. Analysis of their distribution reveals that fewer genes are transcribed at ZT16-20.
*Figure 2. Continued on next page*

*Figure 2. Continued*

(**E**) and (**F**): Rhythmic Nascent-Seq signal was detected for many precursors of non-coding RNAs such as pri-miRNA (d, *pri-miR122a*) and long non-coding RNA(e, lin-ncRNAs *BC019819*, *AK157581*, *BC049268*, *BC056646*).
The following source data and figure supplements are available for figure 2.
**Source data 1.** Gene expression values for all UCSC genes from our mouse liver Nascent-Seq dataset
**Figure supplement 1**. Rhythmic transcription of lncRNA ENSMUSG00000098984 in the mouse liver.
**Figure supplement 2**. Rhythmic transcription of lncRNA ENSMUSG00000086813 in the mouse liver.
**Figure supplement 3**. Rhythmic transcription of lncRNA ENSMUSG00000086771 in the mouse liver.
**Figure supplement 4**. Rhythmic transcription of pri-miRNA pri-Mir17hg in the mouse liver.
**Figure supplement 5**. Rhythmic transcription of pri-miRNA ENSMUSG00000077856 in the mouse liver.
**Figure supplement 6**. Rhythmic transcription of pri-miRNA ENSMUSG00000093077 in the mouse liver.

RNA expression is also observed for other categories, including the AR-AR set (data not shown), suggesting that it is a common feature of transcription vs mRNA comparisons independent of circadian considerations.

Yet a higher variability of transcription (higher SD) correlates with rhythmic mRNA expression within all arrhythmically transcribed genes (n = 4632), suggesting that transcriptional variability generally contributes to the generation of rhythmic mRNA expression (*Figure 6D*). This relationship is independent of nascent RNA expression levels, indicating that the correlation is not due to sequencing depth (*Figure 6E*). More variable transcription is also associated with higher amplitudes of rhythmic mRNA expression (*Figure 6F*). This correlation is valid for most AR-R genes (~80%, *Figure 6F*), suggesting that this transcriptional variability or noise has a significant role in the emergence of rhythmic mRNA expression from arrhythmic transcription.

A comparison between the two replicates of rhythmic mRNAs indicates a better overlap than between one replicate and one replicate of rhythmic nascent RNAs. In contrast, the overlap between the two replicates of rhythmic nascent RNAs was no better than a single replicate rhythmic nascent RNA-rhythmic mRNA comparison (data not shown). Although the two comparisons cannot be definitive because of the limited six time point temporal resolution and resultant noise, they also support more pronounced variation at the transcriptional level than at the mRNA level.

Manual inspection of AR-R genes with high variability revealed a set of genes with high transcription at only one time point (10% of the AR-R gene set; 86/862). This was observed in both replicates and also correlates with rhythmic mRNA expression (*Figure 6G*). mRNA stability (half-life) regulation may contribute to the generation of rhythmic mRNA expression from what is likely a short burst of transcription. This ability of post-transcriptional regulation to generate rhythmic mRNA oscillations is selective, as not all arrhythmically transcribed genes with variable transcription exhibit rhythmic mRNA expression (*Figure 6H*). Moreover, not all genes exhibit variable transcriptional profiles (*Figure 6B*, see below).

About 20% of the AR-R genes are exceptions and exhibit higher variability at the mRNA than at the nascent level (*Figure 6B,F*). Because there is evidence that miRNAs can regulate mRNA levels independently of transcription, we examined whether those genes could be preferentially linked to miRNA regulation. Indeed, a higher number of predicted miRNA target sites was found for these genes compared to genes with higher transcriptional variability (using MirTarget2; p<0.01), suggesting that miRNAs contribute to mRNA cycling of genes with low transcriptional variability (*Figure 6I*).

The separation of these AR-R genes between genes with high or low transcriptional variability is therefore likely to be linked to different modes of gene expression regulation. Interestingly, this separation also reflects distinct biological functions, as AR-R genes with highly variable transcription

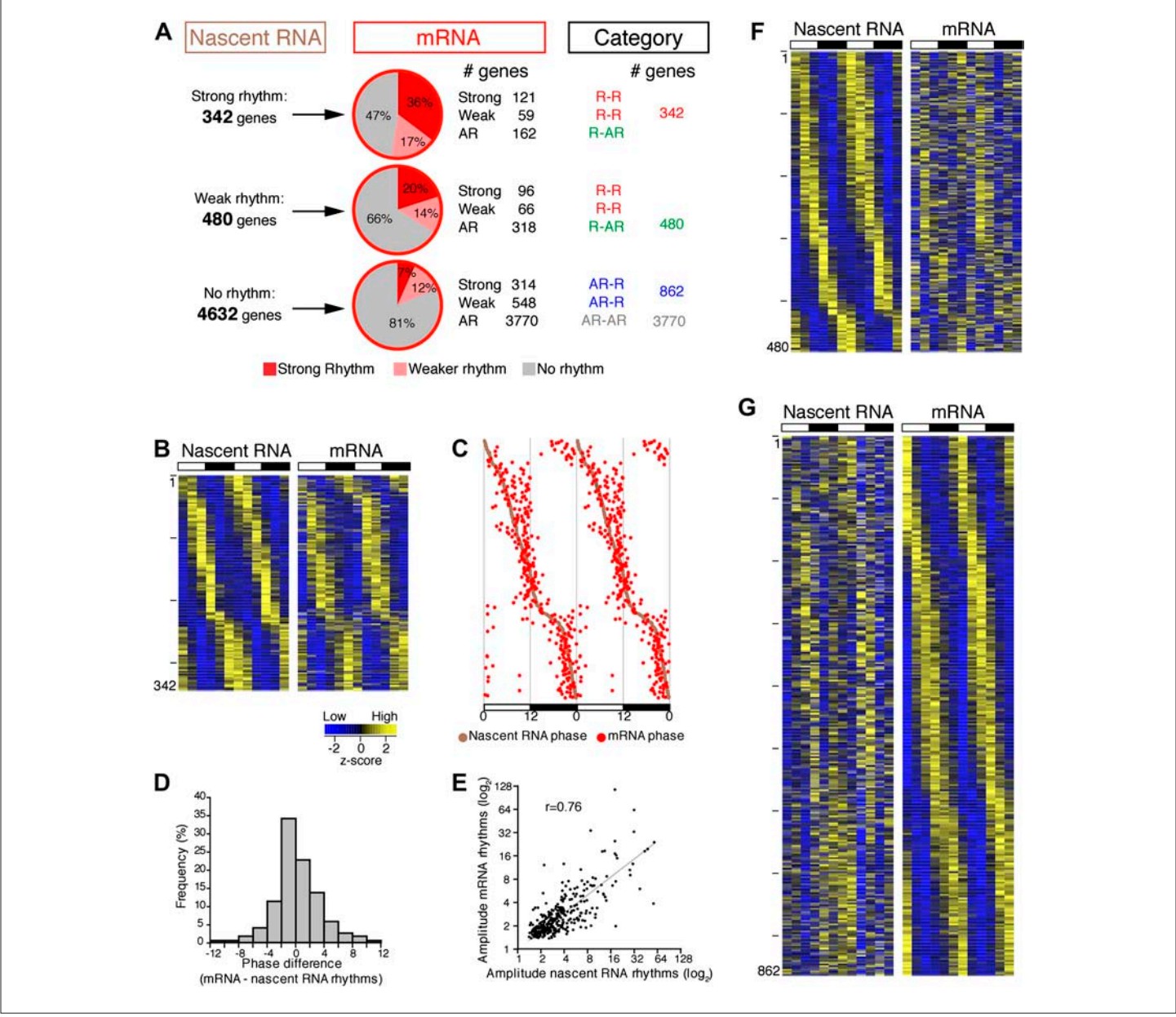

**Figure 3**. Post-transcriptional events account for a significant fraction of rhythmic gene expression in the mouse liver. (**A**): Rhythmic gene expression was assessed as in **Figure 2B** for genes sufficiently expressed in both Nascent-Seq and RNA-Seq datasets. Four categories of rhythmically expressed genes were determined by comparing the Nascent-Seq and RNA-Seq datasets: rhythmic nascent RNA and mRNA (R-R), rhythmic nascent RNA only (R-AR), rhythmic mRNA only (AR-R) and arrhythmic nascent RNA and mRNA (AR-AR). (**B**): Heatmap representation of genes with rhythmic nascent RNA and mRNA expression (n = 342). Classification is based on the phase of nascent RNA oscillations, and each lane corresponds to one gene. (**C**): Double-plotted phase distribution of rhythmic nascent RNA expression (brown) and rhythmic mRNA expression (red) for genes of the R-R gene set. Both phases are highly correlated (r = 0.92). (**D**): Distribution of the difference between the phase of mRNA expression rhythm and the phase of nascent RNA expression rhythm for the 342 R-R genes. (**E**): Amplitude of mRNA expression rhythms are correlated with nascent RNA expression rhythms (r = 0.76). (**F**) and (**G**): Similar representation to (**B**) for rhythmically transcribed genes with no mRNA expression rhythms (**C**, n = 480), and genes that exhibit mRNA oscillations but no rhythms of transcription (**D**, n = 862). For all three heatmaps, high expression is displayed in yellow (z-score > 1), low expression in blue (z-score < 1).

The following source data are available for figure 3.

**Source data 1.** Gene expression values from our Nascent-Seq and RNA-Seq dataset

**Source data 2.** Gene expression values for all UCSC genes from our mouse liver RNA-Seq dataset

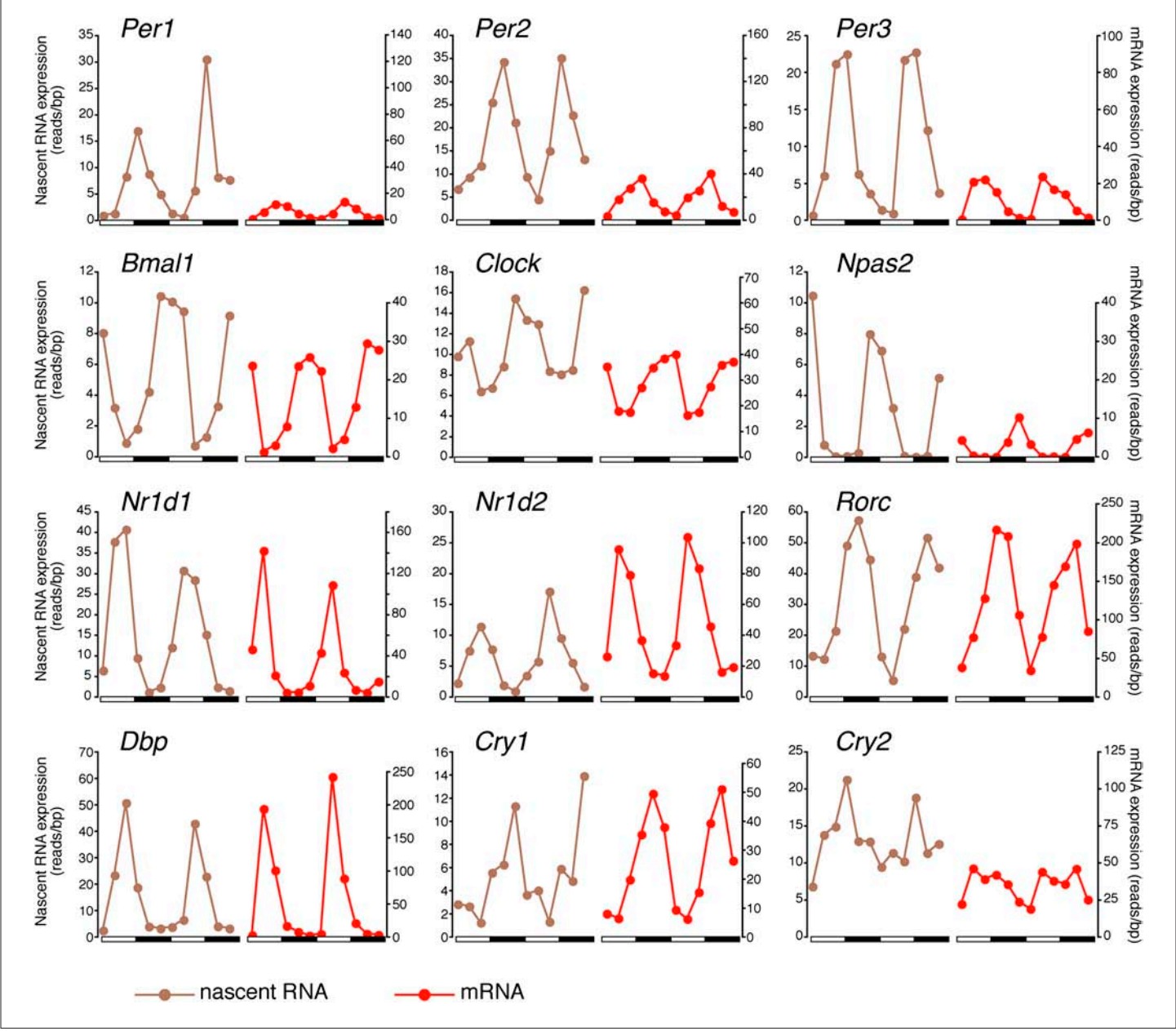

**Figure 4**. Clock genes nascent RNA and mRNA expression in the mouse liver. Clock genes nascent RNA levels (brown; time points every 4 hr starting at ZT0) and mRNA levels (red; time points every 4 hr starting at ZT2) from the Nascent-Seq and RNA-Seq datasets. Relative levels between nascent RNA and mRNA expression profiles are identical for all genes to allow direct comparison.

were enriched for responsive genes (GO: response to stimulus) and metabolic genes (*Figure 6J*), indicating that their intrinsic transcriptional responsiveness is linked to their variable transcriptional profiles.

## Regulation of transcription by CLK:BMAL1

Genes within the R-R gene set include clock genes and many well-characterized clock controlled genes (see above). Because a large fraction of them are directly targeted by the core clock, we asked how CLK:BMAL1 regulates the transcription of its target genes at the genome-wide level. We also took advantage of our Nascent-Seq dataset to establish whether the phase differences between rhythmic

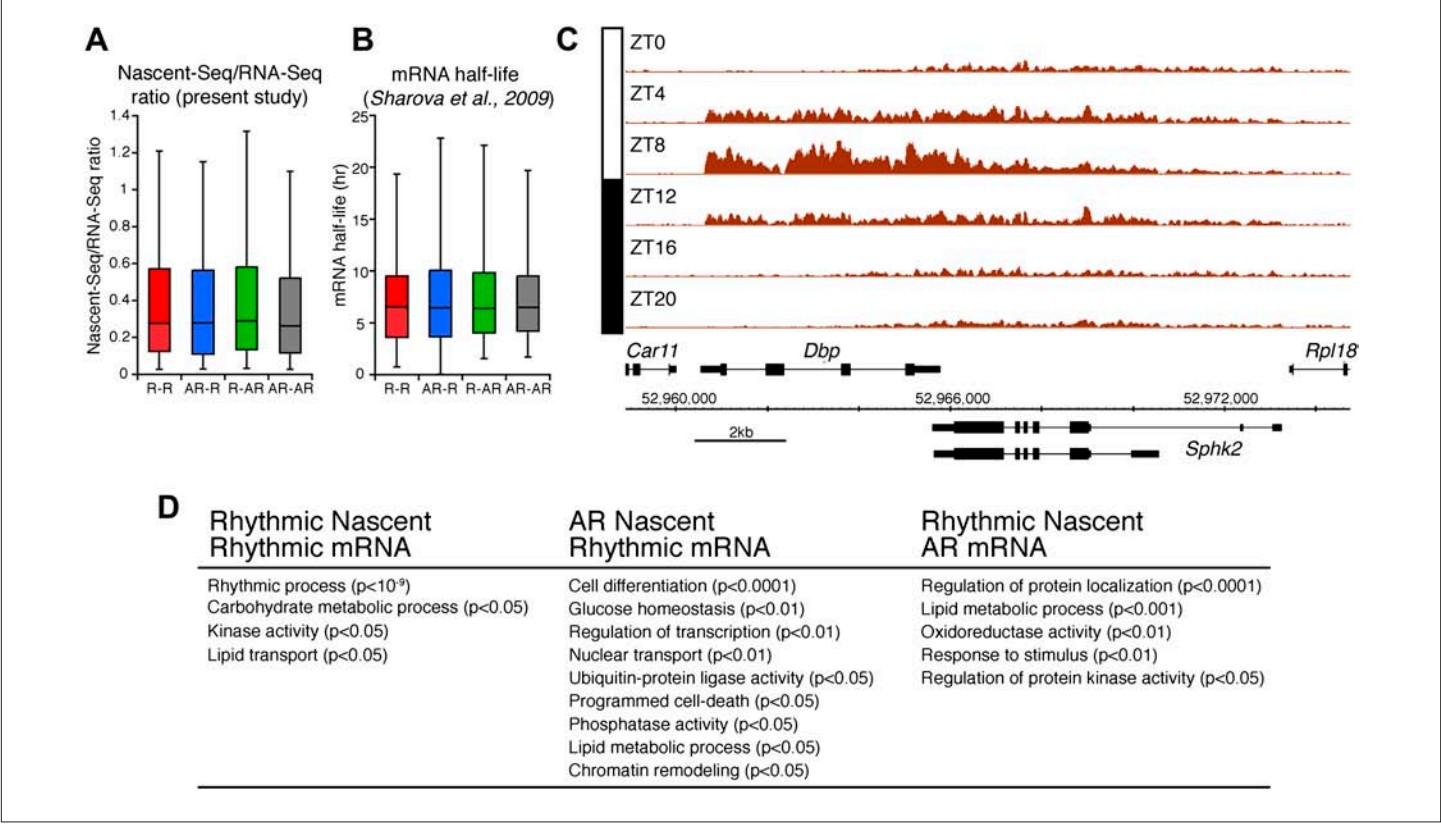

**Figure 5**. Analysis of the different classes of rhythmically expressed genes in the mouse liver. (**A**): Nascent-Seq/RNA-Seq signal ratio (used as inferred half-life) is similar for the four categories of rhythmically expressed genes: rhythmic nascent RNA and mRNA (R-R), rhythmic nascent RNA only (R-AR), rhythmic mRNA only (AR-R) and arrhythmic nascent RNA and mRNA (AR-AR). (**B**): Similar as (**A**), using the RNA half-life values from *Sharova et al., 2009*. (**C**): Nascent-Seq rhythms of 25 of the 480 R-AR genes can be attributed to the rhythmic transcription of an adjacent gene. This applies to *Sphk2* Nascent-Seq rhythm, which likely results from rhythmic *Dbp* nascent RNA signal that extend the 3'end of *Dbp* gene and read through *Sphk2*. Genes above the scale bar are transcribed from left to right and those below the scale bar are transcribed from right to left. (**D**): Gene ontology of three categories of rhythmically expressed genes: rhythmic nascent RNA and mRNA (R-R), rhythmic nascent RNA only (R-AR), rhythmic mRNA only (AR-R).

CLK:BMAL1 DNA binding and rhythmic target gene mRNA expression (*Rey et al., 2011*) reflect transcriptional or post-transcriptional regulation.

To this end, we first performed a ChIP-Seq analysis of CLK and BMAL1 at a time of high DNA binding (ZT8). As expected, CLK and BMAL1 target many DNA binding sites in mouse liver (759 and 1579, respectively) and significantly overlap on 211 of these peaks (*Figure 7A*). Although highly significant (chi-square test, p<0.0001), the rather low fraction may indicate competition between CLK:BMAL1 and NPAS2:BMAL1 for binding sites. Importantly, about 90% of these 211 peaks have been previously characterized as rhythmic BMAL1 DNA binding sites in mouse liver (*Rey et al., 2011*).

CLK and BMAL1 sites overlap at their peak center, consistent with binding to DNA as a heterodimer, and they are enriched for the canonical consensus sequence CACGTG (*Figure 7B,C*) The data therefore indicate that almost all of the 184 direct target genes identified by the 211 CLK:BMAL1 DNA binding sites (*Figure 6—source data 1*) are bona fide direct target genes. They include the expected core clock genes (*Figure 7E–G*) as well as other interesting targets. There are for example 12 ncRNA genes, which include a cluster of four rhythmically transcribed ncRNAs (*Figure 7H*). These cycling ncRNAs suggest novel mechanisms by which CLK:BMAL1 impact circadian rhythms.

Although CLK:BMAL1 target genes are significantly enriched for rhythmically transcribed genes (chi-square test, p<0.0001; *Figure 8A,C*), there is a large discrepancy between the phases of rhythmic BMAL1 DNA binding and those of rhythmic transcription. This is because BMAL1 binding is essentially uniform at ZT3-5, whereas the transcription peaks are much more broadly distributed (*Figure 8A,B*).

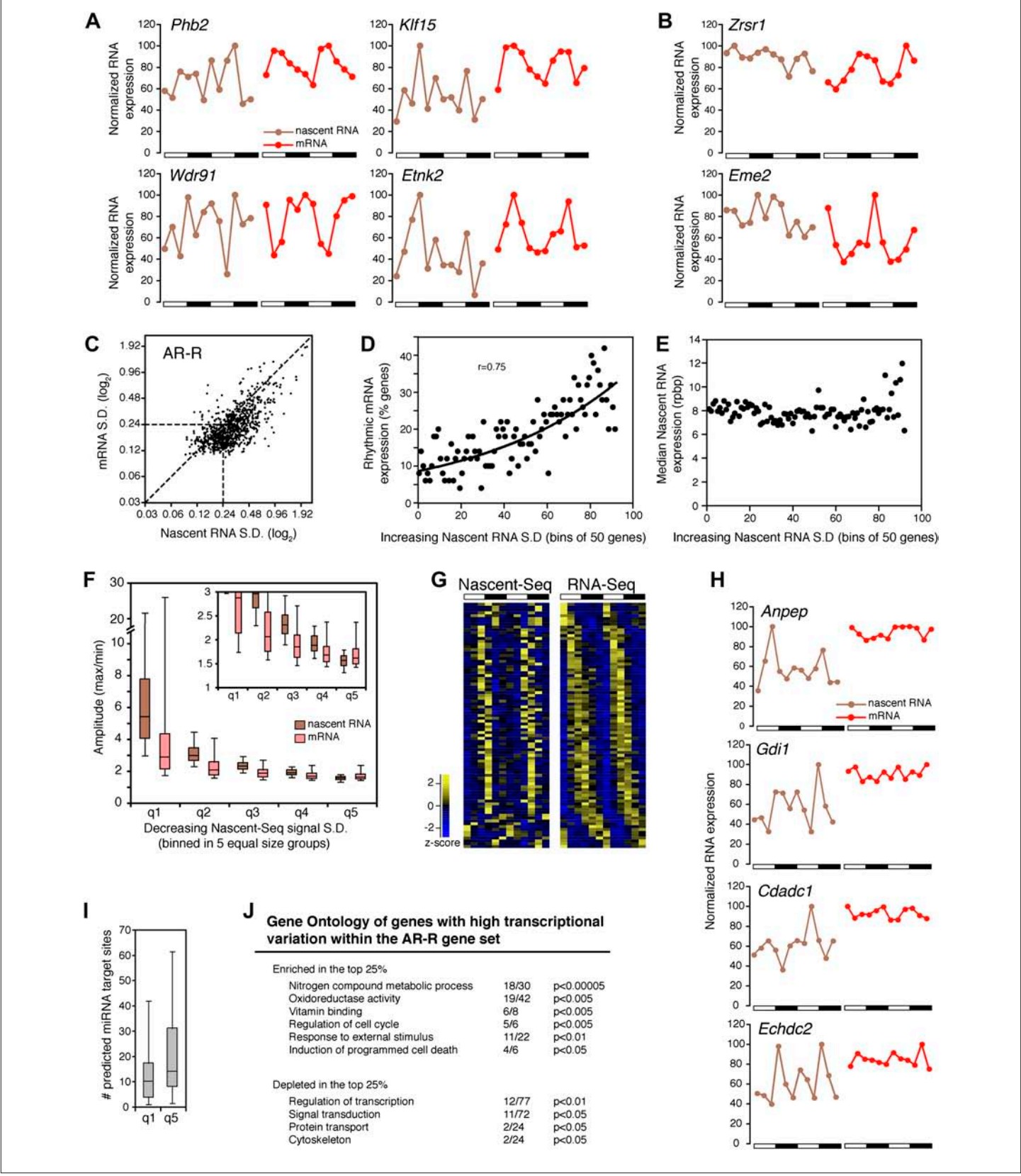

**Figure 6**. Transcriptional variability of AR-R genes contributes to rhythmic mRNA expression. (**A**) and (**B**): Nascent RNA levels (brown; time points every 4 hr starting at ZT0) and mRNA levels (red; time points every 4 hr starting at ZT2) from the Nascent-Seq and RNA-Seq datasets for six genes of the AR-R

*Figure 6. Continued on next page*

*Figure 6. Continued*

gene set. While the majority of the AR-R genes exhibit variable nascent RNA expression (**A**), some of them exhibit a relatively constant transcription when compared to mRNA expression (**B**). (**C**): Standard deviation (SD; calculated using the 12 time points and normalized to the mean) of nascent RNA expression is higher than the SD normalized to the mean of mRNA levels for most AR-R genes. (**D**) and (**E**): Higher transcriptional variability (SD) of arrhythmically transcribed genes is associated with higher occurrence of rhythmic mRNA expression (**D**), but not to nascent RNA expression levels (**E**). (**F**): Higher variability of transcription for the genes of the AR-R group is associated with increase amplitude of rhythms at both Nascent RNA (brown) and mRNA (red) level. Genes of the AR-R group (n = 862) were binned into five quintiles of equal size (q1–q5). (**G**): Heatmap representation of 86 AR-R genes that exhibit high level of transcription at only one time point, and with rhythmic mRNA expression. High expression is displayed in yellow (z-score > 1), low expression in blue (z-score < 1). (**H**): Nascent RNA levels (brown) and mRNA levels (red) for four AR-AR genes with variable nascent RNA expression that is not associated to rhythmic mRNA expression. (**I**): Number of predicted miRNA target sites of AR-R genes with high transcriptional variability (q1, top 20% of the 826 AR-R genes) and low transcriptional variability (q5, bottom 20%). (**J**): Gene ontology of AR-R genes with high transcriptional variability (top 25%) when compared to all AR-R genes. Significant enrichment (top) and depletion (bottom) of biological functions for these genes are displayed. Values correspond to the number of genes within this top 25% of genes, when compared to all AR-R genes.

The following source data are available for figure 6.
**Source data 1.** Peak coordinates for CLK:BMAL1, BMAL1 only and CLK only DNA binding sites

Remarkably, this is also true for the core clock genes. Whereas a small number have the expected phase similar to that of CLK:BMAL1 DNA binding (*Rev-Erbα*, *Dbp*; *Figure 7E*), the transcription of most target genes is significantly delayed. They include *Per1* (*Figure 7F*), *Cry1* (*Figure 7G*) and *Per2* (8E). As recently proposed for *Cry1* (*Ukai-Tadenuma et al., 2011*), this general delay in the peak of rhythmic transcription may be due to a collaboration of CLK:BMAL1 with other transcription factors (*Rev-erbα* and *Dbp* in the case of *Cry 1*).

To validate this interpretation, we assayed the pre-mRNA levels in *Bmal1−/−* mice of four clock genes that exhibit different phases of transcription in wild-type mice (*Figure 8D*). Only the levels of *Rev-Erbα* pre-mRNA in the mutant mice conform to expectation and are at trough levels of wild-type mice. The levels of *Per1*, *Per2* and *Cry1* pre-mRNAs in Bmal1−/− mice are higher than the trough of expression in wild-type mice, indicating that other transcriptional regulators are indeed important for the transcription of these three clock genes (*Figure 8D*). This notion is also supported by the enrichment of other transcription factor motifs such as HNF3/FOXA1, SP1 and E4BP4 adjacent to CLK:BMAL1 binding sites (*Figure 7D*).

Notably, *Per2* has a prominent anti-sense transcript at times of low *Per2* sense transcription (*Figure 8E*). Moreover, the 5′ end of this transcript coincides with peaks of Pol II (*Figure 8E*). This suggests that antisense transcription could be an additional mechanism responsible for the disconnect between the phase of CLK:BMAL1 DNA binding and the phase of rhythmic transcription.

## Discussion

Rhythmic mRNA expression is a hallmark of circadian biology and commonly assumed to be a consequence of rhythmic transcription. However, the application here of Nascent-Seq to genome-wide mouse liver transcriptional rhythms indicates that about 70% of the genes that exhibit rhythmic mRNA expression do not have robust transcriptional rhythms (AR-R category), suggesting that post-transcriptional mechanisms are important for the generation of robust mRNA rhythms. Yet the transcription of most AR-R genes is variable, with elevated levels coinciding with the peak of the rhythmic mRNA profile. This suggests that post-transcriptional events buffer variable transcriptional output to generate robust and reproducible rhythms of mRNA expression (*Figure 9*). A similar Nascent-Seq vs RNA-Seq strategy for *Drosophila* head RNA (Joe Rodriguez and Michael Rosbash, personal communication) and a very recently published paper based on a different strategy for assessing liver transcriptional rhythms (*Koike et al., 2012*) come to a generally similar conclusion, namely, a widespread contribution of post-transcriptional regulation to circadian mRNA cycling.

Relevant mechanisms likely include RNA stability as well as other RNA processing events, as suggested in other systems such as the regulation of gene expression in *Arabidopsis* mitochondria (*Giege et al., 2000*) and after T-cell activation (*Cheadle et al., 2005*). Because the half-lives of nascent transcripts are generally much shorter than those of mRNA (*Griffiths-Jones, 2007*; *Mattick, 2009*; *Mercer et al., 2009*; *Wang and Chang, 2011*), a short burst of transcription can result in elevated mRNA expression

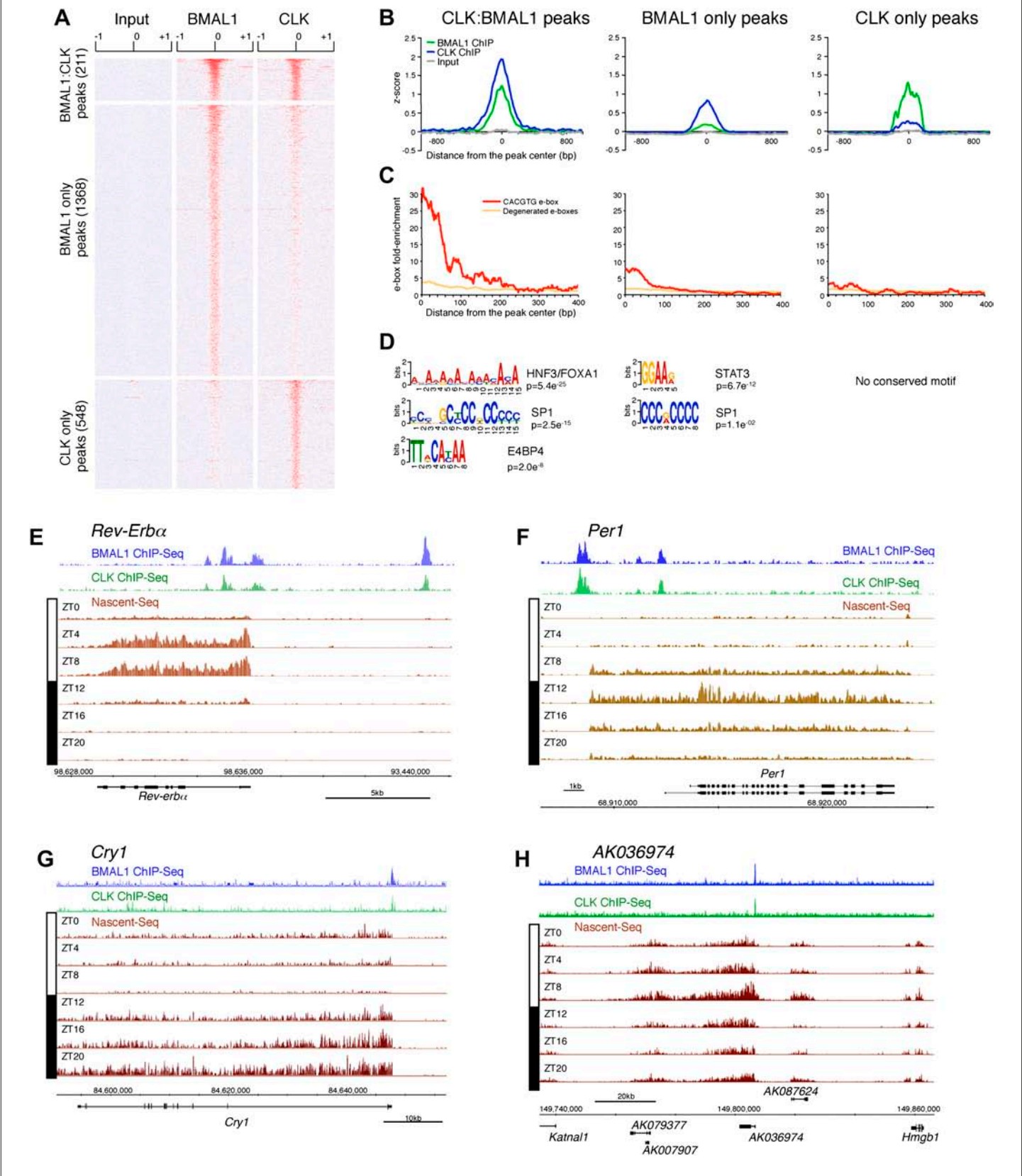

**Figure 7**. Characterization of CLK and BMAL1 target genes in the mouse liver. (**A**) and (**B**): Visualization (**A**) and quantification (**B**) of BMAL1 ChIP-Seq, CLK ChIP-Seq and input signal at BMAL1 and CLK significant peaks (analysis using MACS algorithm). BMAL1 ChIP-Seq, CLK ChIP-Seq and Input signals
*Figure 7. Continued on next page*

*Figure 7. Continued*

were retrieved based on the location of the BMAL1 peaks (center ± 1kb, for CLK:BMAL1 peaks and BMAL1 only peaks) or the CLK peaks (center ± 1kb, for CLK only peaks). Normalization was performed on the entire datasets by calculating the z-score ((x − mean)/SD). Heatmap displays high expression in red and low expression in blue. Quantification (**B**) was performed by averaging the z-score by bins of 25 bp for all CLK:BMAL1 peaks (n = 211), BMAL1 only peaks (n = 1368) and CLK only peaks (n = 548). (**C**): Enrichment of e-boxes (perfect CACGTG in red, degenerated e-boxes [one nucleotide mismatch, in orange]) within ±500 bp of CLK:BMAL1, BMAL1 only and CLK only peak centers. (**D**): Motifs enriched within CLK:BMAL1 peaks, BMAL1 only peaks and CLK only peaks, as revealed by MEME analysis. (**E**)–(**H**): Visualization of BMAL1 ChIP-Seq (blue), CLK ChIP-Seq (green) and Nascent-Seq (brown; six time points of replicate 1) signals for *Rev-Erbα* (**E**), *Per1* (**F**), *Cry1* (**G**) and a cluster of 4 lncRNA (*AK079377, AK007907, AK036974, AK087624*) (**H**) targeted by CLK:BMAL1. Genes above the scale bar are transcribed from left to right and those below the scale bar are transcribed from right to left.

that lasts several hours, as has been shown in systems involving an acute inflammatory response (*Cheadle et al., 2005*; *Hao and Baltimore, 2009*). However, generic mRNA stability cannot account for this buffering, as many genes with arrhythmic but variable transcription do not exhibit rhythmic mRNA expression (*Figure 6H*). This suggests that the post-transcriptional buffering is clock-controlled and selective for specific genes.

Because our experiments were done under LD conditions, it is possible that some cycling RNAs and mechanisms are not circadian but driven by the LD cycle. Nonetheless, it is likely that many of them also occur under DD conditions and that specific and perhaps multiple post-transcriptional mechanisms contribute to rhythmic mRNA expression. They may include 3′ end formation and coupled polyade-nylation, splicing, mRNA export as well as cytoplasmic events involving translation, RNA binding proteins (RBP) and miRNAs (*Joshi et al., 2012*). Interestingly, a few recent reports highlight the tight coupling between transcriptional regulation and post-transcriptional events that govern mRNA stability (*Bregman et al., 2011*; *Trcek et al., 2011*). In these examples, RBPs are recruited by specific transcription factors, which then help load the RBPs onto nascent RNA; they then control cytoplasmic mRNA stability (*Bregman et al., 2011*; *Trcek et al., 2011*). A mechanism of this nature could account for the post-transcriptional generation of rhythmic mRNA expression.

The AR-R genes with high transcriptional variability are enriched for metabolic functions as well as those involved in 'response to stimulus'. The transcription of many metabolic genes is regulated by metabolites and/or hormones (e.g., transcription of Sds is induced by glucagon and CREB, *Haas and Pitot, 1999*). A large fraction of rhythmic mRNAs may therefore result from a transcriptional response, dependent on the cellular environment, as well as post-transcriptional events that stabilize mRNA at an appropriate time of the day. This scenario could also explain the R-AR gene set: despite rhythmic transcription, the lack of rhythmic post-transcriptional regulation would negate the transcriptional oscillations.

Importantly, many genes with rhythmic mRNA expression also exhibit robust transcriptional rhythms. They include all well-known clock genes and many well-characterized clock-controlled genes (see above). Their transcriptional profiles suggest that they are under more stringent transcriptional control (*Figure 9*), due perhaps to direct regulation by the core clock in combination with additional transcription factors.

Our genome-wide characterization of rhythmic transcription also allowed us to directly assay how the rhythmic binding of CLK:BMAL1 to its target gene promoters correlates with transcription. The transcriptional phase of these target genes is heterogeneous and distributed throughout the day, despite a more discrete phase of BMAL1 DNA binding at the beginning of the light phase. This indicates that transcriptional output is not identical for all target genes and suggests that CLK:BMAL1 cooperates with other transcription factors to establish the phase of transcription, as previously shown only for the *Cry1* gene (*Ukai-Tadenuma et al., 2011*). In addition, transcription of most of these target genes is arrhythmic but not absent without BMAL1. This feature of target gene expression as well as the heterogeneity of phase is unlike what is observed in flies: core CLK:CYC target genes exhibit a discrete phase of expression that matches the phase of DNA binding (*Abruzzi et al., 2011*).

In summary, the application of Nascent-Seq and RNA-Seq to mammalian circadian gene expression regulation challenges two assumptions of the mammalian circadian field. The first is that rhythmic transcriptional regulation is sufficient to describe the cycling gene expression landscape. The second is that CLK:BMAL1 DNA binding alone sets the phase of, and is essential for, core clock gene transcription.

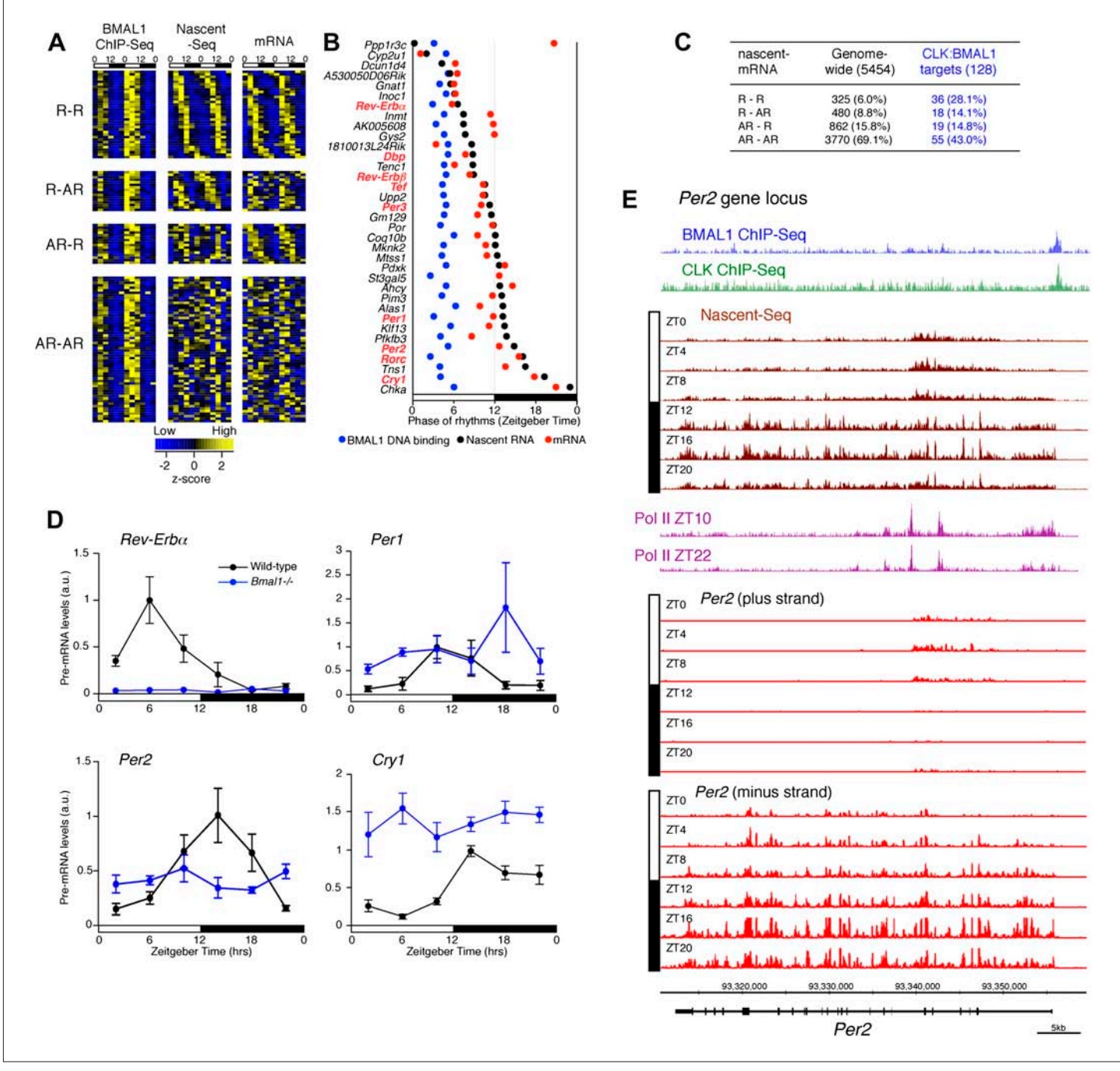

**Figure 8**. Disconnect between rhythmic BMAL1 DNA binding and its transcriptional output. (**A**): Heatmaps representing BMAL1 ChIP-Seq signal (from *Rey et al., 2011*), Nascent-Seq and RNA-Seq signal for CLK:BMAL1 target genes (six time points in duplicate). Genes were classified in four categories: rhythmic nascent RNA and mRNA (R-R), rhythmic nascent RNA only (R-AR), rhythmic mRNA only (AR-R) and arrhythmic nascent RNA and mRNA (AR-AR). High expression is displayed in yellow, low expression in blue. (**B**): Peak phase distribution of rhythmic BMAL1 DNA binding (blue, from *Rey et al., 2011*), of nascent RNA (black) and of mRNA (red) for the direct target genes that are rhythmically expressed at both the nascent RNA and mRNA levels. (**C**): Distribution of CLK:BMAL1 target genes within the 4 different classes of rhythmically expressed genes and its comparison to the genome-wide distribution. Rhythmic nascent RNA and mRNA: R-R; rhythmic nascent RNA only: R-AR; rhythmic mRNA only: AR-R; arrhythmic nascent RNA and mRNA: AR-AR. (**D**): qPCR quantification of *Rev-Erbα*, *Per1*, *Per2* and *Cry1* pre-mRNA every 4 hr throughout the day in wild-type (black, n = 4 per time points) and *Bmal1−/−* mice (blue, n = 3 per time points). Error bar: s.e.m. (**F**): Visualization of BMAL1 ChIP-Seq (blue), CLK ChIP-Seq (green), Nascent-Seq (brown; six time points of replicate 1), Pol II ChIP-Seq signal (purple) at ZT10 and ZT22 (from *Feng et al., 2011*) and strand-specific Nascent-Seq signal for *Per2* (plus strand, top; minus strand, bottom). *Per2* is rhythmically transcribed (minus strand) with a peak at ZT16. An antisense transcript is rhythmically transcribed to *Per2* RNA (plus strand), peaking at ZT4.

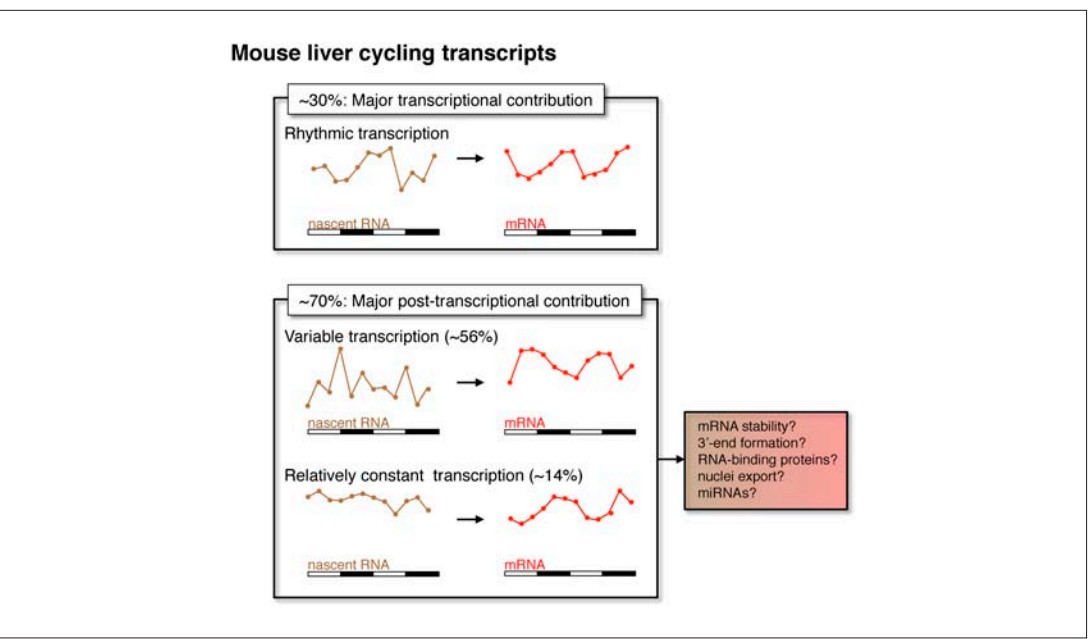

**Figure 9**. Post-transcriptional events contribute to rhythmic mRNA expression in the mouse liver. Although rhythmic transcription plays a major role for approximately 30% of the genes that exhibit rhythmic mRNA expression, post-transcriptional events significantly contribute to the generation of mRNA rhythms for the majority of genes (~70%). Many post-transcriptional cyclers exhibit highly variable transcription that is buffered to generate robust rhythmic mRNA expression. Few genes exhibit a relatively constant transcription when compared to mRNA expression. These post-transcriptional events may include roles for RNA binding proteins and miRNAs to regulate RNA stability, 3′ end formation and nuclei export.

The dramatic, genome-wide disconnect between the phases of rhythmic CLK:BMAL1 DNA binding and rhythmic target gene transcription suggests that other transcription factors and/or mechanisms collaborate with CLK:BMAL1 binding and are critical to determine the phase of clock gene transcription. We anticipate that Nascent-Seq will impact gene expression regulation far beyond the circadian applications shown here.

## Materials and methods

### Animals

3- to 6-month-old male mice housed in a 12 hr-light:12 hr-darkness (LD12:12) schedule were used. Wild-type mice (C57BL/6 strain) and *Bmal1−/−* mice (originally from Christopher A Bradfield; *Bunger et al., 2000*) were used. All experiments were performed in accordance with the National Institutes of Health Guide for the Care and Use of Laboratory Animals and approved by the Brandeis Institutional Animal Care and Use Committee (IACUC protocol #0809-03).

### Isolation of nascent RNA

Mice housed in LD12:12 were sacrificed every 4 hr (ZT0, 4, 8, 12, 16 and 20) by isoflurane anesthesia followed by decapitation. Mouse liver was then quickly removed and homogenized in 3.5 ml of 1× PBS and 3.5 ml of homogenization buffer (2.2 M sucrose, 10 mM Hepes pH 7.6, 15 mM KCl, 2 mM EDTA, 1× protease inhibitor cocktail [Roche, Basel, Switzerland], 0.15 mM spermine, 0.5 M spermidine, 0.5 mM DTT) with a dounce homogenizer (six strokes loose pestle, four strokes tight pestle). The liver homogenate was then mixed with 21.5 ml of homogenization solution and layered on the top of a 10 ml ice-cold cushion solution (2.05 M sucrose, 10 mM Hepes pH 7.6, 10% glycerol, 15 mM KCl, 2 mM EDTA, 1× protease inhibitor cocktail [Roche, Basel, Switzerland], 0.15 mM spermine, 0.5 M spermidine, 0.5 mM DTT) and centrifuged for 45 min at 2°C at 24,000 rpm (100,000×*g*) using a Bechmann SW27 rotor. Nuclei were resuspended in 1 ml of 20 mM Hepes pH 7.6, 150 mM NaCl, 2 mM EDTA, 1× protease

inhibitor cocktail (Roche, Basel, Switzerland), 0.5 mM PMSF, 1 mM DTT, 0.5 U/ml of RNAseOUT/SUPERase-In (Invitrogen/Ambion), homogenized using a 1 ml dounce homogenizer (three times with loose pestle, two times with tight pestle), and divided into three samples of equal volume. One volume of 2× NUN buffer (50 mM Hepes pH 7.6, 2 M Urea, 2% NP-40, 600 mM NaCl, 2 mM DTT, 1× protease inhibitor cocktail [Roche, Basel, Switzerland], 0.5 mM PMSF, 0.5 U/ml of SUPERase-In [Ambion, Carlsbad, California]) was then added drop-by-drop while gently vortexing (level 2). Samples were left on ice for 20 min, then centrifuged at 24,000 rpm for 10 min at 4°C. The supernatant was removed and 1 ml of Trizol (Invitrogen, Carlsbad, California) was added to the DNA pellet. Samples were then incubated at 65°C for 15 min, and DNA pellet was then resuspended by gentle pipetting. Nascent RNA was then extracted using standard Trizol RNA extraction (Invitrogen, Carlsbad, California).

## Isolation of total RNA

Mice housed in LD12:12 were sacrificed by isoflurane anesthesia followed by decapitation every 4 hr (ZT2, 6, 10, 14, 18 and 22). Mouse liver was then quickly removed and cut into small pieces that were frozen on dry ice. Total RNA was extracted using standard Trizol extraction (Invitrogen, Carlsbad, California).

## Generation of Nascent-Seq and mRNA-Seq libraries

Nascent RNA was first DNase-treated with TURBO DNase (Ambion, Carlsbad, California) using manufacturer's recommendation. Polyadenylated RNA was then removed from the nascent RNA using Dynabeads mRNA direct kit (Invitrogen, Carlsbad, California) following manufacturer's recommendations, and nascent RNA was precipitated (ethanol precipitation).

Sequencing libraries have been made using standard protocols. Briefly, 100 ng of purified nascent RNA were used to generate Illumina libraries. Nascent RNA was first fragmented using Fragmentation reagents (AM8740; Ambion, Carlsbad, California) by heating at 70°C for 5 min. Fragmented nascent RNA were then purified and used for standard Illumina library preparation. Following adaptor ligation, libraries of 200–300 bp length were size-selected on a 2% TAE agarose gel, and amplified by PCR for 15 cycles.

Strand-specific libraries were processed as above except for the following modifications that have also been described elsewhere (*Levin et al., 2010*). Briefly, after the first strand cDNA synthesis, dNTPs were removed by size-exclusion chromatography columns (G-50 columns; Amersham, Amersham, UK) and by ethanol precipitation using ammonium acetate. Second strand synthesis was then performed using a dNTP mixture containing dUTP instead of dTTP. After adaptor ligation and size selection (i.e., prior to the amplification), libraries were digested using Uracil-Specific Excision Reagent (USER Enzyme, NEB, Ipswich, Massachusetts) by incubating 2 units (2 µl) of USER Enzyme with 18 µl of libraries at 37°C for 30 min. Reaction was then heat-inactivated, the libraries were purified and PCR-amplified.

RNA-Seq libraries (e.g., mRNA) were made using Truseq RNA sample kit (Illumina, San Diego, California) following manufacturer's recommendations.

## CLK and BMAL1 chromatin immunoprecipitation

Adult mice housed in LD12:12 were sacrificed at ZT8 by isoflurane anesthesia followed by decapitation. Mouse liver was then quickly removed and homogenized in 3.5 ml of 1× PBS supplemented with 1% formaldehyde. After 10 min incubation at room temperature, cross-linking was stopped by mixing liver homogenate with 25 ml of ice-cold quenching solution (2.2 M sucrose, 150 mM glycine, 10 mM Hepes pH 7.6, 15 mM KCl, 2 mM EDTA, 1× protease inhibitor cocktail [Roche, Basel, Switzerland], 0.15 mM spermine, 0.5 M spermidine, 0.5 mM DTT). Homogenate was then layered on top of a 10 ml ice-cold sucrose cushion (2.05 M sucrose, 125 mM glycine, 10 mM Hepes pH 7.6, 10% glycerol, 15 mM KCl, 2 mM EDTA, 1× protease inhibitor cocktail [Roche, Basel, Switzerland], 0.15 mM spermine, 0.5 M spermidine, 0.5 mM DTT) and centrifuged for 30 min at 2°C and 24,000 rpm ($100,000×g$) using a Bechmann SW27 rotor. Nuclei were resuspended in 1 ml of 20 mM Hepes pH 7.6, 150 mM NaCl, 2 mM EDTA and sedimented at $1500×g$ for 1 min. Washed nuclei were resuspended in 1.2 ml sonication buffer (20 mM Hepes pH 7.6, 1% SDS, 150 mM NaCl, 2 mM EDTA) and sonicated on ice using a Fisherbrand Sonic Dismembranator at setting 2 (57 W) for 5 × 15 s to obtain chromatin fragments of about 500–1000 bp in length. The resulting chromatin was centrifuged at $15,000×g$ for 10 min and the resulting supernatant was aliquoted in 200 µl samples for immunoprecipitation and 25 µl samples for input.

Immunoprecipitation of chromatin was performed by mixing 200 µl of sonicated chromatin with 1.8 ml of IP buffer (10 mM Hepes pH 7.6, 150 mM NaCl, 2 mM EDTA, 0.1% NaDeoxycholate, 1% Triton X-100). Antibodies were added and samples were incubated overnight (Rabbit anti-BMAL1 antibody: 10 µl, ab3350; Abcam; Rabbit anti-CLK antibody: 10 µl, NB100-126; Novus Biologicals). Dynabeads protein G (100 µl per sample; Invitrogen, Carlsbad, California) were blocked in parallel overnight in 0.1 mg/ml yeast tRNA and 1 mg/ml BSA in IP buffer.

Following the overnight incubation, beads were washed once with IP buffer. The chromatin/antibodies mixture was then added to the beads and incubated at 4°C for an additional 2 hr. Beads were then washed once for 10 min with HSE I (0.1% SDS, 1% Triton X-100, 2 mM EDTA, 20 mM Hepes KOH, pH 7.6, 150 mM NaCl) and twice for 10 min with HSE II (0.1% SDS, 1% Triton X-100, 2 mM EDTA, 20 mM Hepes KOH, pH 7.6, 500 mM NaCl). Beads were then briefly washed with ice-cold TE and eluted with 200 µl of ChIP Elution buffer (50 mM Tris–HCl pH 8.0, 10 mM EDTA, 1% SDS, 1 mM DTT). 175 µl of ChIP Elution buffer was also added to the 25 µl input samples. Elution was performed at 65°C for 6–18 hr.

The resulting supernatant was removed, supplemented with 200 µl of TE and 8 µl of 1 mg/ml RNAse A (Ambion Cat #2271) and incubated at 37°C for 30 min. Then, 4 µl of 10 mg/ml proteinase K was added and samples were incubated at 55°C for another 2 hr. DNA was then isolated with PCR purification kit (Qiagen) and eluted with 40 µl of elution buffer.

## Generation of Illumina ChIP-seq libraries

BMAL1, CLK and input libraries were made from ChIPs performed from the same mouse liver extract. ChIP-seq libraries were made as described by *Schmidt et al. (2009)*. Size-selected libraries of 200–300 bp length were used for Illumina deep-sequencing, whereas libraries with a 300–650 bp length were used for qPCR validation of the quality of the ChIP-seq libraries.

## High-throughput sequencing of Illumina libraries

High-throughput sequencing has been performed as follow:

- ChIP-seq libraries: BMAL1, CLK and Input ChIP-seq libraries were sequenced using an Illumina Genome Analyzer (GAII) with a sequencing length of 36 nt. To increase depth coverage, libraries were sequenced on multiple lanes (BMAL1: four lanes, CLK: five lanes and Input: three lanes).
- Nascent-Seq libraries: libraries (12 samples corresponding to two independent six-time points rhythms) were sequenced using an Illumina Genome Analyzer (GAII) with a sequencing length of 80 nt. Both replicates of the ZT8 and ZT20 time points were sequenced on two lanes and all other samples on one lane.
- Nascent-Seq libraries, strand-specific: libraries (six samples corresponding to the first replicate of the six-time points rhythm) were generated using bar-coded adaptors, mixed in an equimolar ratio and sequenced on two lanes using a HiSeq2000 (Illumina) with a sequencing length of 101 nt.
- RNA-Seq libraries: libraries (12 samples corresponding to two independent six-time points rhythms) were generated using bar-coded adaptors, mixed in an equimolar ratio and sequenced on two lanes using a HiSeq2000 (Illumina) with a sequencing length of 101 nt.

High-throughput sequencing has been performed following manufacturer recommendations and 8–12 pmol of libraries were hybridized to each lane of the flow-cells. Data were extracted and processed following Illumina recommendations. Sequences were aligned to the mouse genome (UCSC version mm9 database). Number of the sequences obtained for each library can be found in *Table 1*. Datasets are deposited on the Gene Expression Omnibus database under the accession number GSE36916 (GEO, http://www.ncbi.nlm.nih.gov/geo/query/acc.cgi?acc=GSE36916).

## Analysis of Nascent-Seq and RNA-Seq datasets

### Alignment to the mouse genome (mm9 version)

Sequences (fastq format) were first mapped with tophat (*Trapnell et al., 2009*) using the following criteria: -m 1 -g 1 --microexonsearch --no-closure-search -I 500000 (command line: tophat -m 1 -F 0 -g 1 --microexon-search --no-closure-search -G ../exon20110528mm.gtf --phred64-quals -I 500000 -o ZT4_2_tophat.out /data/analysis/fasta/mm9 /data/sequence/Nascent_ZT4_2.fastq). About 65–70% of the Nascent-Seq sequences uniquely mapped to the mouse genome, even

**Table 1.** Number of sequences and statistics for the different sequencing datasets

| | Index number | Barcode | Number of sequences (fastq file) | Number of uniquely mapped sequences | Percentage of uniquely mapped sequences | Normaliz. factor |
|---|---|---|---|---|---|---|
| ChIP-Seq libraries | | | | | | |
| Input | — | — | 39,214,696 | 18,846,303 | 48.1% | — |
| CLK | — | — | 75,944,495 | 37,371,047 | 49.2% | — |
| BMAL1 | — | — | 60,952,293 | 28,920,754 | 47.5% | — |
| Nascent-Seq libraries | | | | | | |
| | | | | | | Norm. 40 m |
| Rep1_ZT0 | — | — | 27,845,320 | 18,319,011 | 65.8% | 2.184 |
| Rep1_ZT4 | — | — | 30,088,981 | 20,931,038 | 69.6% | 1.911 |
| Rep1_ZT8 | — | — | 57,719,174 | 39,567,609 | 68.6% | 1.011 |
| Rep1_ZT12 | — | — | 29,442,244 | 19,485,102 | 66.2% | 2.053 |
| Rep1_ZT16 | — | — | 27,645,102 | 18,385,668 | 66.5% | 2.176 |
| Rep1_ZT20 | — | — | 50,331,242 | 34,703,727 | 69.0% | 1.152 |
| Rep2_ZT0 | — | — | 30,243,856 | 21,014,087 | 69.5% | 1.903 |
| Rep2_ZT4 | — | — | 30,162,514 | 21,082,498 | 69.9% | 1.897 |
| Rep2_ZT8 | — | — | 51,471,477 | 36,118,068 | 70.2% | 1.107 |
| Rep2_ZT12 | — | — | 27,304,921 | 17,815,971 | 65.3% | 2.245 |
| Rep2_ZT16 | — | — | 27,196,805 | 19,077,433 | 70.2% | 2.097 |
| Rep2_ZT20 | — | — | 51,105,236 | 33,547,439 | 65.7% | 1.192 |
| RNA-Seq libraries | | | | | | |
| | | | | | | Norm. 40 m |
| Rep1_ZT2 | 2 | CGATGT | 13,031,496 | 8,693,555 | 66.7% | 4.601 |
| Rep1_ZT6 | 4 | TGACCA | 13,197,078 | 10,214,580 | 77.4% | 3.916 |
| Rep1_ZT10 | 5 | ACAGTG | 13,479,636 | 9,916,774 | 73.6% | 4.034 |
| Rep1_ZT14 | 6 | GCCAAT | 10,366,702 | 7,497,386 | 72.3% | 5.335 |
| Rep1_ZT18 | 7 | CAGATC | 13,147,649 | 9,600,125 | 73.0% | 4.167 |
| Rep1_ZT22 | 12 | CTTGTA | 11,182,756 | 8,233,815 | 73.6% | 4.858 |
| Rep2_ZT2 | 13 | AGTCAA | 14,645,263 | 9,876,359 | 67.4% | 4.050 |
| Rep2_ZT6 | 14 | AGTTCC | 15,836,013 | 12,270,338 | 77.5% | 3.260 |
| Rep2_ZT10 | 15 | ATGTCA | 15,123,726 | 11,507,856 | 76.1% | 3.476 |
| Rep2_ZT14 | 16 | CCGTCC | 12,127,102 | 8,594,609 | 70.9% | 4.654 |
| Rep2_ZT18 | 18 | GTCCGC | 12,903,678 | 9,512,765 | 73.7% | 4.205 |
| Rep2_ZT22 | 19 | GTGAAA | 13,438,873 | 9,592,404 | 71.4% | 4.170 |
| Strand-specific Nascent-Seq libraries | | | | | | |
| | | | | | | Norm. 40 m |
| Rep1_ZT0 | 2 | CGATGT | 34,386,622 | 15,930,801 | 46.3% | 2.511 |
| Rep1_ZT4 | 4 | TGACCA | 45,356,906 | 24,224,151 | 53.4% | 1.651 |
| Rep1_ZT8 | 5 | ACAGTG | 44,309,216 | 24,275,357 | 54.8% | 1.648 |
| Rep1_ZT12 | 6 | GCCAAT | 49,118,104 | 22,882,163 | 46.6% | 1.748 |
| Rep1_ZT16 | 7 | CAGATC | 49,535,738 | 21,835,605 | 44.1% | 1.832 |
| Rep1_ZT20 | 12 | CTTGTA | 54,905,005 | 32,586,396 | 59.4% | 1.228 |

though no rRNA removal has been performed. Uniquely mapped sequences from the tophat output files (bam format) were then used for further analysis. Wig files, used for signal visualization with the Integrated Genome Browser (*Nicol et al., 2009*), were created as described in UCSC website (ftp://hgdownload.cse.ucsc.edu/apache/htdocs-rr/goldenPath/help/bedgraph.html) and normalized to uniquely mapped reads. The normalization factor used for normalization is indicated in the last column of the *Table 1*.

## Quantification of gene signal

Quantification of nascent RNA and mRNA signal has been calculated from the Nascent-Seq and RNA-Seq datasets using custom-written perl and mysql scripts (see *Source code 1*). No quantification was performed using the strand-specific Nascent-Seq datasets. To allow direct comparison between the Nascent-Seq and RNA-Seq datasets, and because many genes have another gene coding for non-coding RNA expressed within their introns (e.g., *Camk2b*, *Figure 2F*), the gene signal was calculated by only considering the sequences mapping to exons. Moreover, because 'only' approximately 13% of the Nascent-Seq signal is mapped to exons (most of the signal is intronic), we reached a higher sequencing depth for the Nascent-Seq samples (>15,000,000 uniquely mapped sequences for all samples, representing a minimum of 160,000,000 nucleotides mapping to the transcriptome).

To allow comparison between the Nascent-Seq and RNA-Seq datasets, individual gene signal was quantitated as previously described (*Rodriguez et al., 2012*) by calculating the number of reads mapping to all exons (regardless of the isoforms) and by normalizing this number to the total exon length (hence the signal was called Reads per Base Pair, or rpbp). This rpbp number was then normalized to the sample sequencing depth (uniquely mapped sequences) and used for further analysis (see below).

Determination of rpbp threshold: the rpbp threshold has been determined so that the difference in gene signal reflects biological variation rather than low sequencing depth. To this end, we took advantage of libraries sequenced on two independent flow-cell lanes and with a similar number of sequences in each lane (e.g., Nascent-Seq, replicate 1, ZT8; 19,207,641 and 20,364,883 sequences respectively).

We first assayed the correlation between gene signals of the two duplicates: the correlation is very good for high values of rpbp, and progressively decreases as the rpbp decreases (*Figure 1G*). The variation of gene signal that results from low sequencing depth was then determined by calculating the standard deviation to the mean (z-score) for every gene between the two duplicates (*Figure 1H*). As expected, the z-score is low for high rpbp and increases with lower gene signal, reflecting a progressive decrease of sequencing depth (*Figure 1H*).

Analysis of this variation allowed us to determine the rpbp threshold. Indeed, among genes with a rpbp superior to 2, less than 6.7% display a variation of more than 1.3-fold between the two duplicates (432/6486) and only 88 genes (1.36%) display a variation of more than 1.5-fold (88/6486; see *Table 2*). Moreover, the majority of these genes are relatively short (<1 kb), hence contributing for some of the variability. This analysis therefore indicates that a rpbp of 2–3 (for datasets with ~20,000,000 uniquely mapped sequences of 80 nt) can be used as a threshold to detect biological variation of nascent RNA expression.

Because of Nascent-Seq datsets were normalized to 40,000,000 reads, we used to threshold of 6 rpbp, as it corresponds to a rpbp of 2.6786 for the Nascent-Seq samples with the least coverage (ZT12, replicate 2). At this threshold, there is very low variation due to sequencing depth (e.g., <4.8% of the genes display a variation of >1.3-fold).

A similar strategy was applied with the RNA-Seq datasets, and set up the threshold as 1–2 rpbp for 10,000,000 sequences (uniquely mapped). This threshold was inferior to the one used with Nascent-Seq datasets, largely because of the superior coverage of exons in the RNA-Seq datasets. Because our analysis was performed on datasets normalized to 40 million reads, we set up the threshold at 8 (which represents a minimum rpbp of 1.5 for the time point with the least coverage; Rep1, ZT14, 7,497,386 sequences).

## Statistics about the levels of expression

Nascent-Seq ZT0, ZT4, ZT12 and ZT16 samples exhibit a relatively similar profile of expression, whereas Nascent-Seq ZT8 and ZT20 samples show higher values because of the higher sequencing depth. For samples with less sequencing coverage, a minimum of 8800 genes have a rpbp >1 (from 8868 for ZT16_rep2 to 10419 for ZT0_rep2), 5100 genes have a rpbp > 2 (from 5121 to 6999) and

**Table 2.** Determination of the rpbp threshold for the Nascent-Seq dataset

| Fold difference | Rpbp > 3 | | Rpbp > 2.6786 | | Rpbp > 2 | |
|---|---|---|---|---|---|---|
| | # Genes | % Genes | # Genes | % Genes | # Genes | % Genes |
| >2 | 0 | 0.00 | 0 | 0.00 | 2 | 0.03 |
| >1.5 | 32 | 0.77 | 44 | 0.93 | 88 | 1.36 |
| >1.4 | 67 | 1.61 | 93 | 1.97 | 180 | 2.78 |
| >1.3 | 169 | 4.06 | 224 | 4.74 | 432 | 6.66 |
| >1.2 | 543 | 13.06 | 689 | 14.57 | 1159 | 17.87 |
| >1.1 | 1698 | 40.83 | 2015 | 42.62 | 3012 | 46.44 |
| 1.0–1.1 | 2461 | 59.17 | 2713 | 57.38 | 3474 | 53.56 |
| Total # genes | 4159 | | 4728 | | 6486 | |

3200 genes have a rpbp > 3 (from 3215 to 4795). More than 1000 genes have a rpbp > 10 in every samples. A file containing the normalized rpbp for all UCSC genes is available in supplementary material (*Figure 2—source data 1*). The normalization to sequencing depth (uniquely mapped sequences) was performed using the normalization factors displayed in the *Table 1*.

All RNA-Seq samples exhibit a relatively similar profile of expression, reflecting the low variation in sequencing depth between the samples. A minimum of 8800 genes have a rpbp >1 (from 8868 for ZT16_rep2 to 10419 for ZT0_rep2), 5100 genes have a rpbp > 2 (from 5121 to 6999) and 3200 genes have a rpbp > 3 (from 3215 to 4795). More than 1000 genes have a rpbp > 10 in every sample. A file containing the normalized rpbp for all UCSC genes is available in supplementary material (*Figure 3—source data 2*). The normalization to sequencing depth (uniquely mapped sequences) was performed using the normalization factors displayed in the *Table 1*.

## Analysis of ChIP-Seq datasets

Sequences from the different libraries (fastq format) were first mapped to the mouse genome (version mm9) using bowtie (*Langmead et al., 2009*) with the command line: bowtie –q –a –-best –m 1. Only those that mapped uniquely to the mouse genome were used for further analysis, and their number has been used for normalization to compare signal difference between libraries.

ChIP-seq libraries were analyzed with the MACS algorithm (*Zhang et al., 2008*) by comparing the treatment sample (BMAL1 or CLK ChIP) to the control sample (Input) using the following criteria: effective genome size = $1.89 \times 10^9$, tag size = 36, band width = 80, model fold = 5, p-value cutoff = $1 \times 10^{-5}$. Significant peaks were computationally assigned to a gene. Briefly, a peak located between the transcription start site and the transcription start end of a gene was assigned to that gene, regardless of the ChIP-Seq peak position. The other peaks, referred as intergenic, were assigned to the gene with the closest transcription start site. Confirmation of this computational gene assignment was then confirmed by manual inspection for the 211 CLK:BMAL1 peaks. Visualization of the ChIP-seq signal was performed using the wig output file (from the MACS analysis) and the IGB browser.

Overlap between CLK and BMAL1 DNA binding peaks has been determined computationally using all significant peaks coordinates. Any overlap between the two peaks (even of one nucleotide) was considered significant. Quantification of the signal has been extracted from the raw data (number of reads per bp) normalized to sequencing depth of each library. For most experiments (e.g., *Figure 7A,B*), signal was binned using a 25 bp window.

Quantification of the number of e-boxes within BMAL1 and CLK DNA binding peaks has been performed computationally using peak fasta sequences. Enrichment has been calculated using the number of e-boxes found at a fixed position from the peak center divided by the expected number of e-boxes. The maximal window size (difference from the fixed position and the peak center) was 500 nt, as the number of expected e-boxes dropped to the background at this window size.

Motif analysis has been performed using MEME suite (http://meme.sdsc.edu/meme/intro.html), using a 100 bp sequence for each peak (peak center ± 50 bp). Parameters were as follow: -dna -mod

anr -nmotifs 20 -minw 6 -maxw 30. The background model contained the same nucleotide distribution as the input file. Significant motifs were then analyzed using TOMTOM from MEME suite.

## Gene ontology of genes with high and low Nascent-Seq/RNA-Seq signal ratio

Gene signal (reads per base pair) was averaged for the 12 time points, and the ratio Nascent RNA/mRNA was calculated. Genes with a ratio over 2 SD from the average of all ratios were selected for gene ontology (GO) analysis; 302 genes had a Nascent-Seq/RNA-Seq ratio below 2 SD and 463 genes had a ratio over 2 SD. 13595 genes were considered for this analysis. GO analysis has been performed using GOToolBox (*Martin et al., 2004*) (http://genome.crg.es/GOToolBox/), using an hypergeometric test with Benjamini-Hochberg correction. The background model consisted of the entire list of genes.

## Statistical analysis of rhythmic gene expression

Only genes with more than three reads per base pair for at least one time point of the Nascent-Seq dataset and two read per base pair for the RNA-Seq dataset were further considered for subsequent analysis (see above). Rhythmically expressed genes were determined based on three parameters: amplitude, F24 (24-hr spectral power, see below) and p-value of the F24.

The amplitude was calculated by dividing the highest value of the 12 time points by the lowest value. The F24 score was calculated by Fourier transformation using a R code originally described by *Wijnen et al. (2005)*. Briefly, normalized reads per base pair from the two independent six-time points rhythms were concatenated and the 24-hr spectral power (F24) was determined for each gene. The F24 score (expressed in range 0–1) indicates the relative strength of the extracted rhythmic component.

The F24 p-value (pF24) represents the probability of observing an F24 score from randomly permuted data that is of equal or greater strength than the extracted Fourier component. It was calculated for each gene after performing 10,000 randomized permutations of the rpbp values. A pF24 was considered significant if (pF24 < 0.05) if the experimental pF24 was within the top 5% of the 10,000 pF24 calculated from randomized permutation.

Transcripts were considered to be rhythmically expressed when meeting the three following criteria: (1) pF24-values < 0.05 (i.e., experimental pF24-value is within the top 5% of the 10,000 pF24-values calculated from randomized permutation), (2) F24 > 0.45 and (3) amplitude (maximal/minimal experimental values) > 1.5. A more stringent cut-off was also used to identify strong rhythmic expression: pF24 < 0.05, F24 > 0.6 and amplitude > 1.75.

The phase information from the Fourier transformation (which indicates the peak of the cosine curve) was further used to assess phase difference between rhythmic nascent RNA and mRNA expression (*Figures 3C,4B*).

## Analysis of gene expression by real-time PCR

Total RNA from wild-type and *Bmal1−/−* mice was prepared from mouse liver using Trizol reagent (Invitrogen) and DNAse-treated using Turbo DNAse (Ambion) according to the manufacturer's protocols. cDNA derived from RNA (using Invitrogen Superscript II and random primers) was utilized as a template for quantitative real-time PCR performed with the Rotor-Gene 3000 real-time cycler (Qiagen). The PCR mixture contained Platinum Taq polymerase (Invitrogen), optimized concentrations of Sybr-green (Invitrogen) and specific primers for either *Rev-Erbα*, *Per1*, *Per2* or *Cry1* pre-mRNAs. Quantitative PCR using *Actg1*-specific primers was used as a loading control. Cycling parameters were 95°C for 3 min, followed by 40 cycles of 95°C for 30 s, 55°C for 45 s, and 72°C for 45 s. Fluorescence intensities were plotted vs the number of cycles by using an algorithm provided by the manufacturer. mRNA levels were quantified using a calibration curve based upon dilution of concentrated cDNA.

## Acknowledgements

We thank Chris Bradfield (University of Wisconsin-Madison, United States) for providing the *Bmal11−/−* mice and the Brandeis University Transgenic Animal Facility for help at various stages of this project. Yevgenia Khodor helped develop the Nascent-Seq technique, and Cindy He and Andrew Darnell provided technical help. We also thank Aoife McMahon and other members of the Rosbash lab as well as Michael Marr, Nelson Lau, Sebastian Kadener and Christine Merlin for helpful discussions and comments on the manuscript.

## Additional information

### Funding

| Funder | Grant reference number | Author |
|--------|------------------------|--------|
| Howard Hughes Medical Institute | | Michael Rosbash |
| National Science Foundation: Integrative Graduate Education and Research Traineeship | DGE 0549390 | Joseph Rodriguez |
| National Institutes of Health: Genetics Training Grant | 5T32GM007122 | Joseph Rodriguez |

The funders had no role in study design, data collection and interpretation, or the decision to submit the work for publication.

### Author contributions

JSM, Performed the experiments, Conception and design, Acquisition of data, Analysis and interpretation of data, Drafting or revising the article; JR, Conception and design, Analysis and interpretation of data, Drafting or revising the article; KCA, Conception and design, Drafting or revising the article; MR, Conception and design, Analysis and interpretation of data, Drafting or revising the article

### Ethics

Animal experimentation: All experiments were performed in accordance with the National Institutes of Health Guide for the Care and Use of Laboratory Animals and approved by the Brandeis Institutional Animal Care and Use Committee (IACUC protocol #0809-03).

## Additional files

### Source code 1.

• Source code 1. Perl script used to calculate gene signal as reads per base pair

### Major datasets

The following datasets were generated:

| Author(s) | Year | Dataset title | Dataset ID and/or URL | Database, license, and accessibility information |
|-----------|------|---------------|------------------------|--------------------------------------------------|
| Menet JS | 2012 | Nascent-Seq Reveals Novel Features of Mouse Circadian Transcriptional Regulation | http://www.ncbi.nlm.nih.gov/geo/query/acc.cgi?acc=GSE36916 | In the public domain at GEO: http://www.ncbi.nlm.nih.gov/geo/ |

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
