## [Decision Letter]

Thank you for choosing to send your work entitled “Nascent-Seq Reveals Novel Features of Mouse Circadian Transcriptional Regulation” for consideration at *eLife*. Your article has been evaluated by a Senior Editor and 3 reviewers, one of whom is a member of *eLife's* Board of Reviewing Editors.

The Senior Editor (Detlef Weigel), the Reviewing Editor (Todd Mockler) and the other reviewers discussed their comments before we reached this decision, and the Reviewing Editor has assembled the following comments based on the reviewers' reports. Our goal is to provide the essential revision requirements as a single set of instructions, so that you have a clear view of the revisions that are necessary for us to publish your work.

**General assessment:**

Your data suggest that the circadian oscillation of mRNA transcription is only partly controlled by the transcriptional regulatory machinery, which is in contrast to what is currently accepted in the field. Your manuscript therefore addresses an important problem in circadian biology: what is the correlation between transcription and the mRNA levels measured by high-throughput techniques? In various organisms and tissues the cycling transcripts have been identified, however, the link between these stable mRNA measures, transcriptional activity, and post-transcriptional regulation has not been addressed on a global scale. The reviewers agreed that the principles behind the experiments are generally sound, that the manuscript is well written, and that with a few exceptions the experimental protocols and analysis methods are clear. This paper provides several key advancements in the field and provides a solid foundation upon which to build a better understanding of the transcriptional regulatory mechanisms and the contribution of post-transcriptional mechanisms to clock regulated gene expression.

**Major concerns:**

Several concerns relate to establishing whether the findings arising from the Nascent-Seq and mRNA-seq comparisons reflect genuine biological differences or are a technical artifact from comparing two different methods:

It is difficult to evaluate the conclusions drawn from the nascent Seq, mRNA-seq, and ChIP-seq experiments without knowing how much Illumina read data was acquired for each sample, and what the read mapping statistics were for each sample. Carefully describing the amount and types of sequence data is essential for interpretation and needs to be presented in the manuscript so that the reader can confirm that the conclusions are based on genuine biological phenomena. Also, you used strand-specific libraries for some of the experiments, but it is not clear in which ones, and if this could affect the analysis. Please provide this information, which can be presented in an additional table. How many genes were expressed in each sample (i.e. detected in the sequence data given some threshold of minimal expression) in the nascent Seq and mRNA-seq experiments needs to be given as well, as this will be a further measure of whether the depth of sequencing achieved for each sample was adequate.

A second question is whether the quantification method affect the variance. Is the read per base pair value adjusted for the total number of reads? If not, differences in sequencing depth could contribute to differences in variance. Further, the number of introns and extended regions in the nascent RNA data could affect the proportion of exonic sequence being sampled, biasing the sensitivity of the read per base pair measure. Appropriate normalization would contribute to ensuring the comparisons are as even as possible.

The comparison of mRNA levels in the same tissue from different labs has also resulted in non-overlapping sets of rhythmic transcripts. Is any of the difference in nascent RNA and mRNA levels from intrinsic biological variation or from the limitations of quantifying these molecules? How consistent are the mRNA and nascent RNA timecourses compared to themselves (e.g. the 6 samples of mRNA to the other 6 samples of mRNA). Since this will not provide sufficient resolution, other techniques to establish the similarity of the data sets can be used. What is the overlap with rhythmic data from other labs (using the same algorithms)?

There are several new comparisons and analysis methods described in this paper. It is essential that the approaches and analysis methods employed on this novel data are clearly explained and accessible to the community so that the experiments can be applied to other systems and comparisons with this work made. The description of the analysis methods provided in the materials and methods need to be detailed and where “custom scripts” are used, these scripts need to be made available.

Cycling nascent RNA and mRNA was compared with a time course from the livers of mice housed in 12:12 LD. Interpretation is hindered by use of a cycling light/dark cycle as it is impossible to distinguish which rhythms are truly circadian and which are light driven; collections in constant darkness would have been preferable. While the data presented is still compelling, this potential confounding factor needs to be addressed in a revised manuscript.

Comparison of rhythmic nascent RNA and mRNA (R-R) showed a weak overlap between the two data sets (only 41.6% of na-RNAs were also rhythmic at the mRNA level). However, it is unclear how this percentage was reached as the ratio of 342 (strong rhythmic nascent RNA) to 822 (total rhythmic nascent RNA) was used, but the authors indicate that 342 is representative of “rhythmically transcribed genes” (and suggest all cycling nascent RNA) whereas 842 represents “rhythmic RNA expression” (lines 114-116). Based on the figures provided, it is unclear how these numbers were obtained and further clarification is needed. A 28.4% overlap of oscillating Na-RNA and total mRNA (342/1204) was shown, which suggests a high amount of cycling mRNAs undergo post-transcriptional regulation that is critical for rhythmicity. However, it is unclear why only the strongly rhythmic nascent RNAs were used (342) instead of all cycling Na-RNAs (842) and this should be explained more clearly.

**Summary:**

The major issues the review team would like to see addressed relate to read depth, normalization and comparison methods, and improved descriptions of the methods to better explain how the data was processed and analyzed (for example, better explanations of key ratios used in data interpretation). The results of this study are very exciting, but it is important for the authors to convince the reader that the results reflect a genuine biological difference and do not reflect a technical artifact from comparing two different methods. The review team hopes your manuscript will be able to move forward with improved analysis and better explanation of this analysis, without the need for new data.

---

## [Author Response]

*Several concerns relate to establishing whether the findings arising from the Nascent-Seq and mRNA-seq comparisons reflect genuine biological differences or are a technical artifact from comparing two different methods*:

To address this concern, we have included a detailed description of the methods/analysis used in the manuscript.

*It is difficult to evaluate the conclusions drawn from the nascent Seq, mRNA-seq, and ChIP-seq experiments without knowing how much Illumina read data was acquired for each sample, and what the read mapping statistics were for each sample. Carefully describing the amount and types of sequence data is essential for interpretation and needs to be presented in the manuscript so that the reader can confirm that the conclusions are based on genuine biological phenomena (point #1). Also, you used strand-specific libraries for some of the experiments, but it is not clear in which ones, and if this could affect the analysis. Please provide this information, which can be presented in an additional table (point #2). How many genes were expressed in each sample (i.e. detected in the sequence data given some threshold of minimal expression) in the nascent Seq and mRNA-seq experiments needs to be given as well, as this will be a further measure of whether the depth of sequencing achieved for each sample was adequate (point #3)*.

Point #1: The detailed information about the number of reads acquired for each sample and the read mapping statistics are now provided (see the Materials and methods).

Point #2: We clarified how and when we used the strand-specific data. Briefly, we only used them for visualization purposes, that is to assess whether antisense transcription within some transcription units may be relevant for the Nascent-Seq profile (e.g., *Per2*, see Figure 5F). No quantification/analysis of gene signal has been performed using the strand-specific datasets.

Point #3: The information about the number of genes expressed in each sample is now provided in the expanded Materials and methods. Moreover, we now provide the Nascent-Seq and RNA-Seq signal for every gene and every sample. Because all samples have substantial read coverage, a minimum of 160,000,000 nucleotides per sample mapped to the transcriptome, this corresponds to an average of 2.5 reads/bp for all transcripts.

*A second question is whether the quantification method affect the variance. Is the read per base pair value adjusted for the total number of reads? If not, differences in sequencing depth could contribute to differences in variance (point #1). Further, the number of introns and extended regions in the nascent RNA data could affect the proportion of exonic sequence being sampled, biasing the sensitivity of the read per base pair measure. Appropriate normalization would contribute to ensuring the comparisons are as even as possible (point #2)*.

Point #1: The reads per base pair (rpbp) value was adjusted to the total number of reads. This is now stated and explained in the manuscript.

Point #2: As mentioned in the expanded Materials and methods, we required a higher sequencing depth for the Nascent-Seq data than for the RNA-Seq for this exact reason, i.e., less exonic signal in Nascent-Seq data. Moreover, the rpbp threshold was determined by comparing the rpbp values of a single library sequenced in parallel in two lanes. The comparison is now explained in detail. Briefly, it allowed us to determine a rpbp threshold for which there was minimal variability between the two lanes of sequencing.

For comparisons of rpbp signal between the Nascent-Seq and RNA-Seq data, we only compared the ratio Nascent-Seq rpbp/RNA-Seq rpbp for only the exons of every gene. So this makes intron signal irrelevant except for the requirement for more sequencing depth.

*The comparison of mRNA levels in the same tissue from different labs has also resulted in non-overlapping sets of rhythmic transcripts. Is any of the difference in nascent RNA and mRNA levels from intrinsic biological variation or from the limitations of quantifying these molecules? How consistent are the mRNA and nascent RNA timecourses compared to themselves (e.g. the 6 samples of mRNA to the other 6 samples of mRNA). Since this will not provide sufficient resolution, other techniques to establish the similarity of the data sets can be used. What is the overlap with rhythmic data from other labs (using the same algorithms)*?

We are aware of these important issues. Although they are complicated, our sequencing depth and methodological strategies are covered elsewhere in these answers. What has not been mentioned, however, are the robust and highly reproducible rhythmic profiles of the clock genes. This argues against major and systematic methodological problems as the sole cause of transcriptional variability, consistent with bona fide variability for some genes and not for others (a conclusion of our paper).

With respect to data from other labs, the non-overlapping set of mRNAs found by different lab also results to some extent from the use of different statistical methods to analyze the data. Our study uses the same pipeline for the entire analysis of rhythmic gene expression. Other possible reasons are differences in mouse strains, how animals are raised, their manipulation, the food, etc. Because all our experiments were done with the same mice in the same facility and performed by the same experimenter (J.S.M), the mRNA vs Nascent RNA comparisons cannot suffer from these issues.

In response to the reviewers’ suggestion, we performed a comparison between the two independent replicates. Although the analysis cannot be completely meaningful/quantitative because of the limited six time point temporal resolution and resultant noise (as anticipated by the reviewers), it shows better overlap between the two RNA-Seq datasets than between a single RNA-Seq and a single Nascent-Seq dataset. In contrast, the overlap between the two Nascent- Seq datasets was no better than the Nascent-Seq/RNA-Seq data comparison, supporting more pronounced variation at the transcriptional level than at the mRNA level. This is indeed a major conclusion reached in our manuscript. A statement summarizing these new comparisons (thanks to this reviewer for the suggestion) has been added to the paper.

The other suggested comparison is limited by the lack of any published RNA-Seq dataset that address rhythmic gene expression in the mouse liver. However, we did carefully compare our data with an existing microarray dataset (e.g., Kornmann et al., 2006). We used our analytical methods to identify cycling mRNAs in the dataset. We chose this study because it used the same paradigm, e.g., 2 independent 6-time points rhythms in LD. Although the comparison is of limited value because of the many technical differences, i.e., quality of the microarray probes, different probes for one single gene that result in different rhythmic output etc, those cycling mRNAs match better our RNA-Seq dataset than our Nascent-Seq dataset, consistent again with the results reported in the manuscript.

*There are several new comparisons and analysis methods described in this paper. It is essential that the approaches and analysis methods employed on this novel data are clearly explained and accessible to the community so that the experiments can be applied to other systems and comparisons with this work made. The description of the analysis methods provided in the materials and methods need to be detailed and where “custom scripts” are used, these scripts need to be made available*.

The new Materials and methods section should solve this problem. Importantly, the scripts used to calculate RNA expression levels are available as a supplementary file.

*Cycling nascent RNA and mRNA was compared with a time course from the livers of mice housed in 12:12 LD. Interpretation is hindered by use of a cycling light/dark cycle as it is impossible to distinguish which rhythms are truly circadian and which are light driven; collections in constant darkness would have been preferable. While the data presented is still compelling, this potential confounding factor needs to be addressed in a revised manuscript*.

We have made sure this issue is clarified in the revised manuscript. However, we suspect that only a minority of genes will be different between LD and DD in mammalian liver. Note: this is based on the current understanding of entrainment physiology.

*Comparison of rhythmic nascent RNA and mRNA (R-R) showed a weak overlap between the two data sets (only 41.6% of na-RNAs were also rhythmic at the mRNA level). However, it is unclear how this percentage was reached as the ratio of 342 (strong rhythmic nascent RNA) to 822 (total rhythmic nascent RNA) was used, but the authors indicate that 342 is representative of “rhythmically transcribed genes” (and suggest all cycling nascent RNA) whereas 842 represents “rhythmic RNA expression” (lines 114-116). Based on the figures provided, it is unclear how these numbers were obtained and further clarification is needed. A 28.4% overlap of oscillating Na-RNA and total mRNA (342/1204) was shown, which suggests a high amount of cycling mRNAs undergo post- transcriptional regulation that is critical for rhythmicity. However, it is unclear why only the strongly rhythmic nascent RNAs were used (342) instead of all cycling Na-RNAs (842) and this should be explained more clearly*.

This issue has been clarified in the revised version of the manuscript. It includes all these numbers mentioned above by the reviewers. It also looks like this confusion comes from the number 342, which is the number of genes with strong rhythmic Nascent-Seq expression (as shown in Figure 3A) but coincidentally also the number of genes in the R-R category (as described in the text).